# Citalopram exhibits immune-dependent anti-tumor effects by modulating C5aR1[+] TAMs

Fangyuan Dong[1†], Shan Zhang[2†], Kaiyuan Song[3†], Luju Jiang[2†], Li-Peng Hu[2], Qing Li[2], Xue-Li Zhang[2], Jun Li[2], Mingxuan Feng[4], Zhi-Wei Cai[5], Hong-Fei Yao[5], Rong-Kun Li[6], Hui Li[2], Jie Chen[1], Xiaona Hu[1], Jiaofeng Wang[1], Chongyi Jiang[5], Helen He Zhu[7], Cun Wang[2], Lin-Tai Da[3], Zhi-Gang Zhang[2*], Zhijun Bao[1*], Xu Wang[6*], Shu-Heng Jiang[2*]

[1]Department of Gastroenterology, Shanghai Key Laboratory of Clinical Geriatric Medicine, Shanghai Institute of Geriatrics and Gerontology, Huadong Hospital, Fudan University, Shanghai, China; [2]State Key Laboratory of Systems Medicine for Cancer, Shanghai Cancer Institute, Ren Ji Hospital, School of Medicine, Shanghai Jiao Tong University, Shanghai, China; [3]Key Laboratory of Systems Biomedicine (Ministry of Education), Shanghai Center for Systems Biomedicine, Shanghai Jiao Tong University, Shanghai, China; [4]Department of Liver Surgery, Ren Ji Hospital, School of Medicine, Shanghai Jiao Tong University, Shanghai, China; [5]Department of General Surgery, Hepato-biliary-pancreatic Center, Huadong Hospital, Fudan University, Shanghai, China; [6]Institute of Oncology, Affiliated Hospital of Jiangsu University, Zhenjiang, China; [7]State Key Laboratory of Systems Medicine for Cancer, Renji-Med-X Stem Cell Research Center, Shanghai Cancer Institute and Department of Urology, Ren Ji Hospital, Shanghai Jiao Tong University School of Medicine, Shanghai, China

*For correspondence:
zzhang@shsci.org (Z-GZ);
zhijunbao@fudan.edu.cn (ZB);
wangxu@ujs.edu.cn (XW);
shjiang@shsci.org (S-HJ)

[†]These authors contributed equally to this work

Competing interest: The authors declare that no competing interests exist.

## eLife Assessment

This **important** study provides **solid** evidence to support the anti-tumor potential of citalopram, originally an anti-depression drug, in hepatocellular carcinoma (HCC). In addition to their previous report on directly targeting tumor cells via glucose transporter 1 (GLUT1), the authors tried to uncover additional working mechanisms of citalopram in HCC treatment in the current study. The data here suggests that citalopram may regulate the phagocytotic function of TAM via C5aR1 or CD8+T cell function to suppress HCC growth in vivo.

**Abstract** Administration of selective serotonin reuptake inhibitors (SSRIs) is associated with a reduced cancer risk and shows significant anti-tumor effects across multiple tumor types, suggesting the potential for repurposing SSRIs in cancer therapy. Nonetheless, the specific molecular target and mechanism of action of SSRIs remain to be fully elucidated. Here, we reveal that citalopram exerts an immune-dependent anti-tumor effect in hepatocellular carcinoma (HCC). Interestingly, the anti-HCC effects of citalopram are not reliant on its conventional target, the serotonin transporter. Through various drug repurposing approaches, including global reverse gene expression profiling, drug affinity responsive target stability assay, and molecular docking, the complement component 5a receptor 1 (C5aR1) is identified as a new target of citalopram. C5aR1 is predominantly expressed by tumor-associated macrophages, and citalopram treatment enhances local macrophage phagocytosis and elicits CD8[+] T anti-tumor immunity. C5aR1 deficiency or depletion of CD8[+] T cells hinders the anti-HCC effects of citalopram. Collectively, our study reveals the immunomodulatory roles of

citalopram in inducing anti-tumor immunity and provides a basis for considering the repurposing of SSRIs as promising anticancer agents for HCC treatment.

## Introduction

The development of anti-tumor drugs encounters numerous challenges, such as a high failure rate, substantial costs, limited bioavailability, safety concerns, and lengthy design and testing procedures (*Pushpakom et al., 2019*). As different diseases may share common pathological mechanisms, there is a growing focus on repurposing existing drugs for new applications, referred to as 'new uses for old drugs' (*Foretz et al., 2023*; *Schipper et al., 2022*). Additionally, exploring new disease-relevant targets for established drugs and broadening their indications offers significant translational value (*Xia et al., 2024*).

Selective serotonin reuptake inhibitors (SSRIs; i.e., citalopram, escitalopram, fluoxetine, fluvox-amine, paroxetine, and sertraline) are frequently prescribed to treat conditions such as depression, anxiety, insomnia, and chronic pain (*Fluyau et al., 2022*). SSRIs target the serotonin transporter (SERT, encoded by the *SLC6A4* gene), blocking the reuptake of serotonin (5-hydroxytryptamine, 5-HT) into the presynaptic neuron, ultimately increasing 5-HT levels in the brain (*Singh et al., 2023*). Recent population-based epidemiological studies suggest that SSRIs have the potential to reduce the risk of many cancers, including kidney, breast, colorectal, and liver cancers (*Bhagavathula et al., 2022*; *Fischer et al., 2022*; *Lee et al., 2021*; *Zhang et al., 2021*). The anxiety and depression symptoms among cancer patients are increasingly recognized (*Carlson et al., 2023*). Considering the established safety profile and known side effects of SSRIs, repurposing these drugs as potential anti-tumor agents, particularly for cancer patients with comorbid depression, could offer dual benefits. This approach may not only help improve neurological function but also potentially impede tumor progression. Hence, repurposing SSRIs could present a cost-effective option. Nevertheless, the mechanisms of action responsible for the anti-tumor effects of SSRIs are still largely unknown.

In this study, using hepatocellular carcinoma (HCC) as a model, we revealed that citalopram, an SSRI, exhibits anti-tumor properties not through its traditional target SERT, but instead via partially immune-dependent mechanisms. Through a comprehensive approach involving global reverse gene expression profiling, drug affinity responsive target stability (DARTS) (*Lomenick et al., 2009*), and molecular docking analyses, we identified the complement component 5a receptor 1 (C5aR1), predominantly expressed by tumor-associated macrophages (TAMs), as a target of citalopram. C5aR1 plays critical roles in fostering an immunosuppressive microenvironment (*Beach et al., 2023*; *Medler et al., 2018*) and boosting the metastatic potential of cancer cells (*Ajona et al., 2018*). Targeting C5aR1[+] TAMs effectively reverses tumor progression and enhances anti-tumor response (*Li et al., 2024*; *Luan et al., 2024*; *Zhao et al., 2024*). Given its significant impact on cancer progression, C5aR1 has surfaced as a promising therapeutic target for potential cancer interventions. Here, we further elucidated the mechanism underlying the connection between C5aR1[+] TAMs and citalopram-related phenotypes.

## Results

### The immune-dependent anti-tumor effects of citalopram in HCC

Previously, we have demonstrated the direct effects of citalopram on cancer cell proliferation, apoptosis, and metabolic processes (*Dong et al., 2024*). To further evaluate the therapeutic effects of citalopram, we engrafted mouse Hepa1-6 and Hep53.4 cells into different strains of recipient mice, including the immune-competent C57BL/6 mice and immune-compromised $Rag1^{-/-}$ mice. Citalopram was given by i.p. administration daily for 15–25 days after bearing palpable tumors (~50 mm³). Tumor growth was significantly retarded in both two mouse models. Of note, this impairment was more pronounced in immune-competent C57BL/6 mice (*Figure 1A, B*), indicating an immune-dependent effect of citalopram, although the possibility of other non-immune mechanisms remains. This finding was further validated at molecular levels, as evidenced by reduced staining of the proliferation index Ki67 and increased expression of cleaved caspase-3 (CCS3) (*Figure 1—figure supplement 1*).

As SERT is the known target of SSRIs, we tested the therapeutic effect of citalopram in the absence of SERT (*Figure 1C, D*). In mouse HCC cells, SERT knockdown did not significantly impact cell proliferation or apoptosis, both in vitro and in vivo (*Figure 1E, F*, *Figure 1—figure supplement 2A, B*).

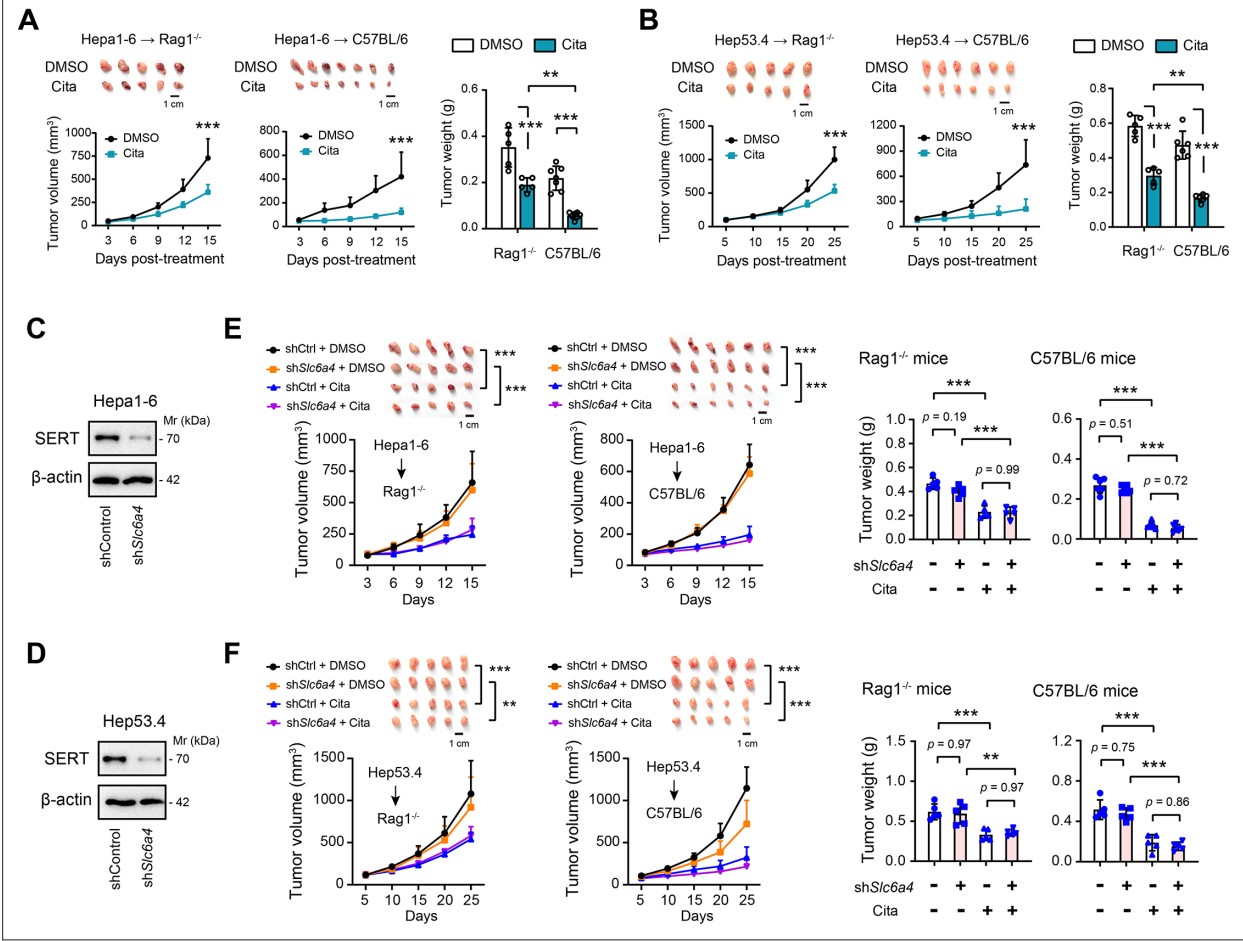

**Figure 1.** The immune-dependent and SERT-independent anti-tumor effects of citalopram in hepatocellular carcinoma (HCC). (**A, B**) Mouse HCC cells, Hepa1-6 or Hep53.4, were subcutaneously injected into $Rag1^{-/-}$ C57BL/6 or immunocompetent C57BL/6 mice ($n$ = 5–7 per group). When bore visible tumors, 5 mg/kg citalopram was treated daily for 15 or 25 days. Tumors were excised after mice were sacrificed, and the tumor weight was measured. (**C, D**) Western blotting showed the knockdown efficiency of SERT in Hepa1-6 and Hep53.4 cells. (**E, F**) shControl and sh$Slc6a4$ Hepa1-6 and Hep53.4 cells were subcutaneously injected into $Rag1^{-/-}$ C57BL/6 or immunocompetent C57BL/6 mice ($n$ = 5–6 per group). When bore visible tumors, 5 mg/kg citalopram was treated daily for 15 or 25 days. Tumors were excised after mice were sacrificed, and the tumor weight was measured. In all panels, *p < 0.05, **p < 0.01, ***p < 0.001. Values are presented as mean ± SD and compared by one-way analysis of variance (ANOVA) multiple comparisons with Tukey's method (for bar chart comparison) and two-way ANOVA with Dunnett's multiple comparisons (for survival curve comparison).

The online version of this article includes the following source data and figure supplement(s) for figure 1:

**Source data 1.** Original western blots for *Figure 1C, D*, indicating the relevant bands.

**Source data 2.** Original files for western blot analysis displayed in *Figure 1C, D*.

**Figure supplement 1.** Citalopram inhibits hepatocellular carcinoma (HCC) cell proliferation and promotes cell apoptosis in the immune-competent and immune-deficient mouse models.

**Figure supplement 2.** Citalopram suppresses hepatocellular carcinoma (HCC) cell proliferation and promotes cell apoptosis in an SERT-independent manner.

However, citalopram delayed the growth of SERT-silenced HCC tumors, and its suppressive effect was more proficient in immune-competent hosts compared with $Rag1^{-/-}$ ones (*Figure 1E, F*, *Figure 1—figure supplement 2C–F*). Together, these data suggest that citalopram might target other molecules instead of SERT to exhibit an immune-dependent anti-tumor effect in HCC. Using a single-cell sequencing dataset of HCC (GSE125449), we revealed that SERT is expressed not only in HCC cells but also in T cells, tumor-associated endothelial cells, and cancer-associated fibroblasts (*Figure 1—figure supplement 2G*). Therefore, we cannot fully rule out the possibility that citalopram may influence these cellular components within the tumor microenvironment (TME) and contribute to its therapeutic effects.

## C5aR1 is a direct target of citalopram

Previously, using drug-induced gene signature networks and target prediction method, we identified a repertoire of targets related to citalopram treatment (*Dong et al., 2024*). The top two hits are glucose transporter 1 (GLUT1) and C5aR1 (*Figure 2A, B*). Citalopram can reverse the Warburg effect to inhibit HCC by targeting GLUT1 directly (*Dong et al., 2024*). Considering the significant role of C5aR1 in immune modulation (*Pio et al., 2019*), we investigated whether the immune-dependent anti-tumor effects of citalopram are mediated through C5aR1. Immunohistochemical analysis showed that the positive staining of C5aR1 was predominantly found in scattered immune cells (*Figure 2C*). Single-cell sequencing data indicated that C5aR1 was expressed at higher levels in monocytes/macrophages within the TME compared with other cell types (*Figure 2D*). Co-immunofluorescence analysis corroborated that CD163+ macrophages presented significant C5aR1 immunoreactivity (*Figure 2E*). As previously reported (*Zhang et al., 2019*), six distinct macrophage subclusters have been identified in HCC, with M4-c1-THBS1 and M4-c2-C1QA demonstrating significant enrichment in tumor tissues. The M4-c1-THBS1 cluster is associated with signatures characteristic of myeloid-derived suppressor cells, whereas the M4-c2-C1QA cluster displays features resembling those of TAMs as well as M1 and M2 macrophages. We found that C5aR1 is highly expressed in these two clusters, while the expression levels in the other macrophage clusters were considerably lower (*Figure 2—figure supplement 1*).

The DARTS assay is a widely recognized technique used to identify potential protein targets for small molecules. This approach involves conferring proteolytic protection of target proteins by their interaction with small molecules. Combined DARTS assay with immunoblotting analysis, we investigated the stability of C5aR1 against pronase in the presence or absence of citalopram treatment. Because C5aR1 is expressed by macrophages, THP-1 cells were employed for the DARTS assay. Citalopram did not affect the basal protein level of C5aR1 in macrophages (*Figure 2F*). However, citalopram could enhance the stability of C5aR1 against pronase treatment. Of note, this effect was specific to C5aR1, as citalopram did not impact the proteolytic susceptibility of β-actin (*Figure 2F*). Moreover, the stability of C5aR1 against pronase was enhanced in a dose-dependent manner following treatment with citalopram (*Figure 2G*). Besides citalopram, other 5 SSRIs (escitalopram, fluoxetine, fluvoxamine, paroxetine, and sertraline) also played protective roles of C5aR1 against pronase (*Figure 2—figure supplement 2A*).

To address citalopram/C5aR1 interaction, in silico docking analysis was conducted. Previous studies revealed two different inhibitor-binding sites in C5aR1, namely, the orthosteric site on the extracellular side and the allosteric site outside the transmembrane bundle (*Robertson et al., 2018*). Therefore, we docked the structure of citalopram into both orthosteric and allosteric sites of human C5aR1 (PDB id: 6C1Q), respectively. In silico docking of citalopram into the orthosteric site predicted two binding conformations without overlap (*Figure 2H*, *Figure 2—figure supplement 2B*). In pose-1, the fluorophenyl and the cyanophtalane group of citalopram stacked with the side chains of Y258 and R206, and the amino group of citalopram interacted with the negatively charged residue E199 (*Figure 2I*, left). In pose-2, two negatively charged residues, D282 and D37, surrounded the amino group (*Figure 2I*, middle). Considering the accessibility, D282 orienting toward the binding cavity of C5aR1 might play a more pivotal role in the citalopram binding event proposed by pose-2. Additionally, the cyanophtalane group of citalopram interacted with hydrophobic residues L92 and P113 of C5aR1. The fluorine was inserted into a narrow space lying over the side chain of W102 (*Figure 2I*, right). For the allosteric site, residue W213 played a critical role in inhibitor binding via hydrogen-bond interactions established by the amino group in the indole ring (*Liu et al., 2018*). According to the best pose from cluster 2, the citalopram formed a hydrogen bond via the oxygen atom in the cyanophtalane group with W213 (*Figure 2I*, right). Inspired by these molecular docking results, we generated six C5aR1 mutants to figure out the real binding site of citalopram in HEK293T cells (*Figure 2—figure supplement 2C*). Indeed, citalopram had no protective effects on C5aR1 stability with the transfection of HA-C5aR1 E199 and D282 mutant against pronase (*Figure 2J*). Consistently, mutations on E199 and D282 reduce C5a binding affinity to C5aR1 (*Wang et al., 2023b*). Furthermore, docking analysis and DARTS assay revealed that the other four SSRIs could bind to the D282 site of C5aR1 (*Figure 2—figure supplement 2D, E*). These findings demonstrate that SSRIs adopt the orthosteric site to bind to C5aR1 and electrostatic interactions between C5aR1 D282 and the amino groups of SSRIs play an instrumental role in the binding mechanisms.

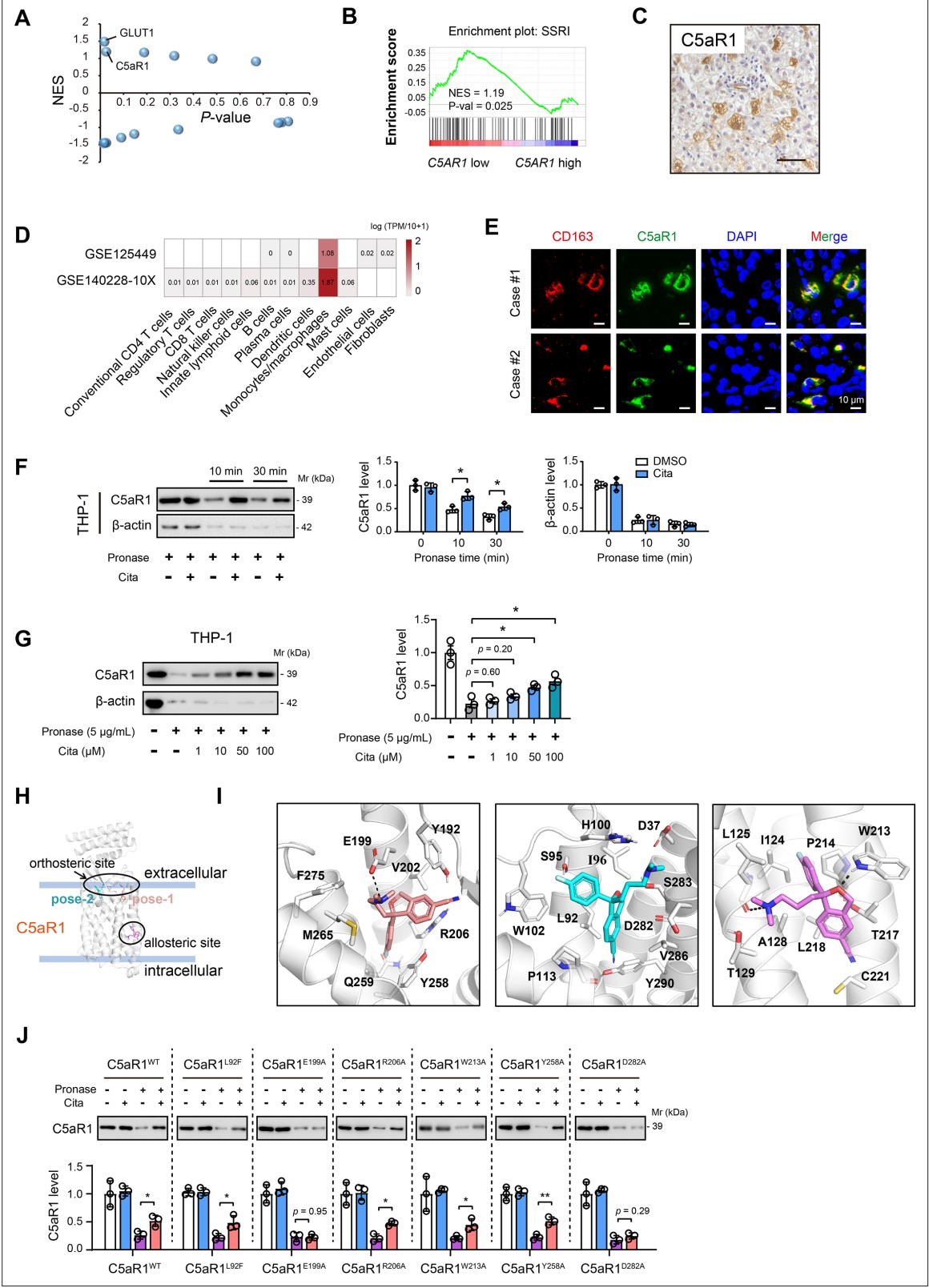

**Figure 2.** C5aR1 is a direct target of citalopram. (**A**) Gene Set Enrichment Analysis (GSEA) identified GLUT1 and C5aR1 as two top hits related to selective serotonin reuptake inhibitor (SSRI)-induced gene changes. (**B**) GSEA of hepatocellular carcinoma (HCC) RNA-seq data (TCGA cohort) with the SSRI-related gene signature. Sample grouping was made based on the median expression of C5aR1. (**C**) Representative immunohistochemical images showed the expression pattern and cellular distribution of C5aR1 in human HCC tissues. Scale bar, 50 μm. (**D**) Single-cell RNA sequencing analysis

*Figure 2 continued*

showed the expression pattern of C5aR1 with the immune microenvironment of HCC. (**E**) Co-immunofluorescence of C5aR1 (green) with CD163 (red) in HCC samples. Scale bar, 10 µm. (**F**) The drug affinity responsive target stability (DARTS) assay and immunoblot analysis showed C5aR1 protein stability against 5 µg/ml pronase in the presence and absence of 100 µM citalopram treatment. (**G**) The DARTS assay and immunoblot analysis showed C5aR1 protein stability against 5 µg/ml pronase in the presence of different concentrations of citalopram treatment. (**H**) The overall conformation of citalopram binding to C5aR1. (**I**) Representative models of citalopram in pose-1 (left), pose-2 (middle), and allosteric site (right). Several polar interactions were indicated by black dashed lines. (**J**) HEK293T cells were transfected with either WT or mutant C5aR1 expression plasmids for 48 hr, followed by DARTS assay with immunoblotting analysis of C5aR1 protein levels. In all panels, *p < 0.05, **p < 0.01. Values as mean ± SD and compared by the Student's *t* test (**F**) or one-way analysis of variance (ANOVA) multiple comparisons with Tukey's method among groups (**G, J**).

The online version of this article includes the following source data and figure supplement(s) for figure 2:

**Source data 1.** Original western blots for *Figure 2F, G, J*, indicating the relevant bands and treatments.

**Source data 2.** Original files for western blot analysis displayed in *Figure 2F, G, J*.

**Figure supplement 1.** The expression pattern of C5aR1 in tumor-associated macrophages (TAMs) in hepatocellular carcinoma (HCC).

**Figure supplement 2.** Identification of C5aR1 as a direct target for citalopram.

**Figure supplement 2—source data 1.** Original western blots for *Figure 2—figure supplement 2A, E*, indicating the relevant bands and treatments.

**Figure supplement 2—source data 2.** Original files for western blot analysis displayed in *Figure 2—figure supplement 2A, E*.

## Citalopram targets C5aR1 in TAMs

C5aR1, the receptor for anaphylatoxin C5a, is profoundly implicated in tumor progression via inducing an immunosuppressive microenvironment in which TAMs are involved (*Medler et al., 2018*). As C5aR1 is localized in TAMs within the TME of HCC, we explored whether citalopram exerted anti-tumor functions by targeting C5aR1 on TAMs. Given that citalopram also targets GLUT1 in HCC (*Dong et al., 2024*), we generated a Hepa1-6 clone with complete knockdown of GLUT1 (GLUT1^KD^) to dissect C5aR1-dependent roles (*Figure 3A*). Interestingly, citalopram displayed an inhibitory effect on GLUT1^KD^ tumors in immune-component C57BL/6 mice but not *Rag1*^−/−^ ones (*Figure 3B*), underscoring the contribution of C5aR1. To delineate the involvement of macrophages, we first depleted mouse macrophages with clodro-Liposomes and found that citalopram was ineffective in controlling GLUT1^KD^ tumors (*Figure 3—figure supplement 1*), implying the influences of citalopram on macrophages. As the second line of evidence, we engrafted GLUT1^KD^ Hepa1-6 cells into the syngeneic *C5ar1*^+/−^ and *C5ar1*^−/−^ mice. Compared with *C5ar1*^+/−^ recipients, tumor growth was attenuated in *C5ar1*^−/−^ recipients (*Figure 3C*), whereas C5a deposition remained unchanged (*Figure 3D*), suggesting that while C5a is still present, its interaction with C5aR1 is critical for influencing tumor growth dynamics. Third, we injected GLUT1^KD^ Hepa1-6 cells into syngeneic recipient *C5ar1*^−/−^ mice that had been reconstituted with donor *C5ar1*^+/−^ or donor *C5ar1*^−/−^ bone marrow (BM) cells to analyze the therapeutic effect of citalopram (*Figure 3E*). Implantation of donor *C5ar1*^+/−^ BM into *C5ar1*^−/−^ recipients restored tumor growth compared with their *C5ar1*^−/−^ counterparts, and citalopram cannot restrict tumor growth in recipient *C5ar1*^−/−^; donor *C5ar1*^−/−^ mice; in contrast, tumor growth kinetics were considerably diminished in recipient *C5ar1*^−/−^; donor *C5ar1*^+/−^ mice upon citalopram treatment (*Figure 3F*), suggesting that C5aR1^+^ TAM rewiring enables the tumoricidal effects of citalopram. Notably, citalopram treatment did not affect C5a deposition (*Figure 3G*), highlighting that the impact of citalopram is primarily on C5a/C5aR1 interactions and downstream signaling pathways, rather than on altering C5a levels.

To answer how C5aR1^+^ macrophages mediate citalopram-induced anti-tumor effects, we probed macrophage functions. Given the crucial role of the C5a/C5aR1 axis in macrophage phagocytosis (*Wang et al., 2023a*), we investigated whether citalopram has an effect on this process. Phagocytosis assay showed that isolated TAMs from *C5ar1*^−/−^ recipients had higher phagocytic capacity of HCC cells than that of *C5ar1*^+/−^ recipients (*Figure 3H*). Consistently, Gene Set Enrichment Analysis (GSEA) indicated a significant enrichment of the phagocytosis pathway in macrophages derived from *C5ar1*^−/−^ mice compared to those from *C5ar1*^+/−^ mice (*Figure 3—figure supplement 2A*). Moreover, citalopram increased the phagocytosis of TAMs from recipient *C5ar1*^−/−^; donor *C5ar1*^+/−^ mice (*Figure 3I*). C5a-mediated macrophage phagocytic impairment was reversed by C5aR1 knockdown or treatment with citalopram or different SSRIs (*Figure 3—figure supplement 2B–D*). C5a was able to suppress, while citalopram was effective in bolstering phagocytosis in macrophages expressing C5aR1^WT^ but not C5aR1^D282A^ (*Figure 3—figure supplement 2E, F*). Despite several residues of mouse C5aR1 protein being different from that of human C5aR1, citalopram shared a similar binding model with them

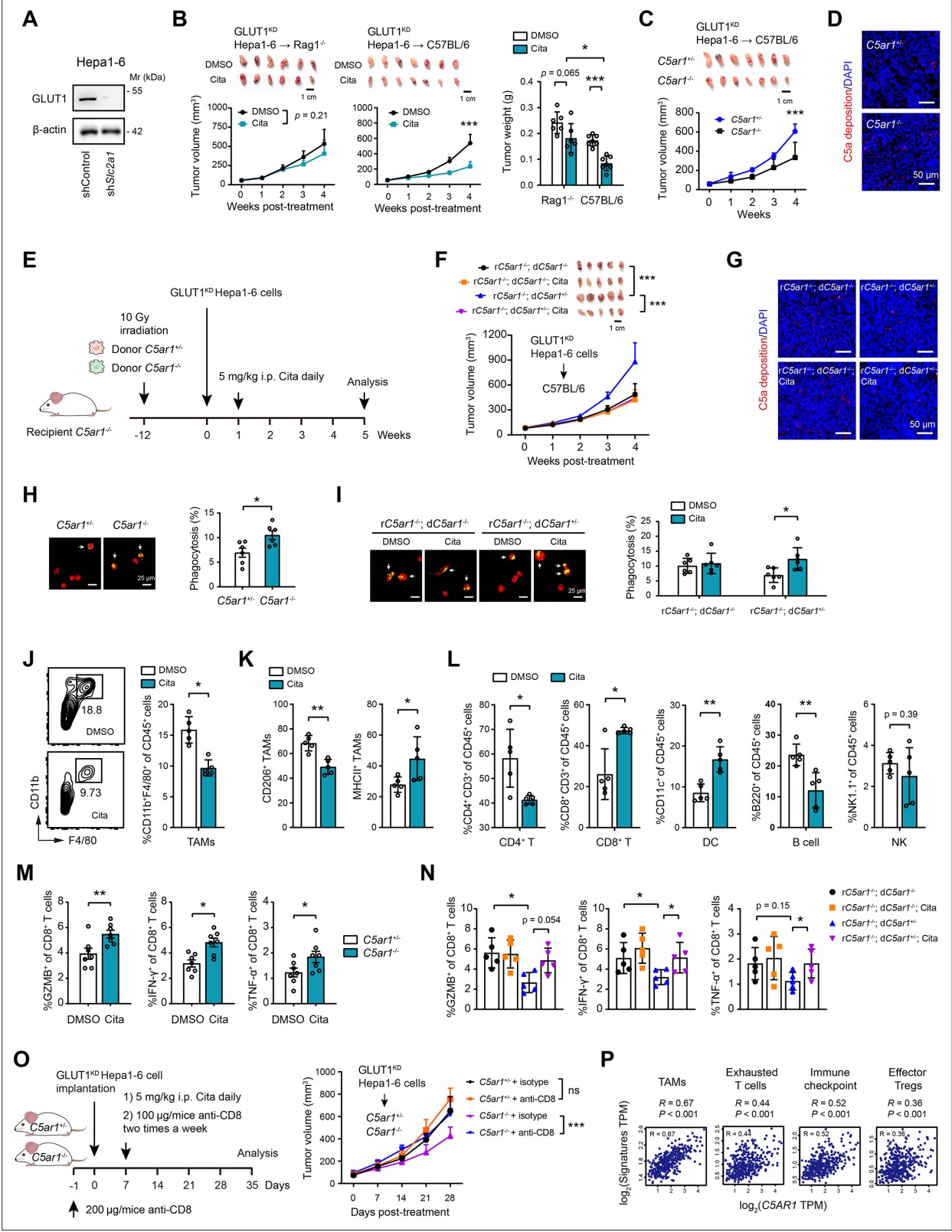

**Figure 3.** Citalopram targets C5aR1+ tumor-associated macrophages (TAMs). (**A**) Western blotting showed the knockdown efficiency of GLUT1 in mouse Hepa1-6 cells. (**B**) GLUT1KD Hepa1-6 cells were subcutaneously injected into the *Rag1−/−* or immunocompetent C57BL/6 mice, and mice were treated with 5 mg/kg citalopram when bore visible tumors; 3 weeks later, tumor burden was examined (*n* = 6–7 per group). (**C**) The growth kinetics of GLUT1KD Hepa1-6 tumors in *C5ar1+/−* and *C5ar1−/−* C57BL/6 host (*n* = 7). (**D**) Immunofluorescence analysis of C5a deposition in GLUT1KD Hepa1-6

*Figure 3 continued*

tumors from *C5ar1*[+/−] and *C5ar1*[−/−] C57BL/6 host. Scale bar, 50 µm. (**E**) Experimental design of bone marrow transfer experiments. (**F, G, I**) GLUT1[KD] Hepa1-6 cells were subcutaneously implanted into syngeneic recipient (r) mice that had been reconstituted with bone marrow cells from either *C5ar1*[+/−] or *C5ar1*[−/−] donor mice. The therapeutic effect of citalopram (**F**), C5a deposition (**G**), and macrophage phagocytosis (**I**) in this model was analyzed. Scale bar, 50 µm. (**H**) The phagocytic capacity of macrophages isolated from GLUT1[KD] Hepa1-6 tumors in *C5ar1*[+/−] and *C5ar1*[−/−] C57BL/6 host. Flow cytometry showed the infiltration of CD45[+]CD11b[+]F4/80[+] macrophages (**J**), CD206[+] TAMs and CD11b[+] TAMs (**K**), tumor-infiltrating lymphocytes (**L**) in tumor tissues from orthotopic xenograft model, which was generated in immunocompetent C57BL/6 mice with Hepa1-6 cells (*n* = 5 per group). (**M, N**) Measurement of CD8[+] T cell function in tumor tissues from the groups mentioned in **C** and **F**. (**O**) The growth kinetics of GLUT1[KD] Hepa1-6 tumors in *C5ar1*[+/−] and *C5ar1*[−/−] C57BL/6 host upon CD8[+] T cell depletion (*n* = 7). (**P**) Correlation analysis of C5aR1 expression and immune checkpoint molecules, gene signatures of TAMs, exhausted T cells, and effector Tregs in the TCGA cohort (*n* = 371). In all panels, *p < 0.05, **p < 0.01, ***p < 0.001; ns, non-significant. Values are presented as mean ± SD and compared by two-way analysis of variance (ANOVA) with Dunnett's multiple comparisons (**B, C, F, O**), Student's *t* test (**H–M**), one-way ANOVA multiple comparisons with Tukey's method (**B, N**), and the Spearman's rank correlation methods (**P**).

The online version of this article includes the following source data and figure supplement(s) for figure 3:

**Source data 1.** Original western blots for *Figure 3A*, indicating the relevant bands.

**Source data 2.** Original files for western blot analysis displayed in *Figure 3A*.

**Figure supplement 1.** Macrophage depletion mitigates the anti-tumor effects of citalopram.

**Figure supplement 2.** Citalopram reverses C5a-mediated macrophage phagocytic impairment via targeting C5aR1.

**Figure supplement 2—source data 1.** Original western blots for *Figure 3—figure supplement 2B, E*, indicating the relevant bands.

**Figure supplement 2—source data 2.** Original files for western blot analysis displayed in *Figure 2—figure supplement 2B, E*.

**Figure supplement 3.** Predicted binding modes of citalopram to human and mouse C5aR1.

**Figure supplement 4.** Expression pattern and cellular functions of GLUT1 and GLUT3 in macrophages.

**Figure supplement 4—source data 1.** Original western blots for *Figure 3—figure supplement 4C*, indicating the relevant bands.

**Figure supplement 4—source data 2.** Original files for western blot analysis displayed in *Figure 3—figure supplement 4C*.

**Figure supplement 5.** Real-time qPCR analysis of macrophage polarization and flow cytometry analysis of immune cells in tumor tissues and spleen.

(*Figure 3—figure supplement 3*). Given citalopram's reported role in inhibiting GLUT1, we tested whether citalopram affects macrophage metabolism. However, the glycolytic metabolism of THP-1 cells remained largely unaffected following citalopram treatment, as evidenced by glucose uptake, lactate release, and extracellular acidification rate (ECAR) (*Figure 3—figure supplement 4A*). Next, we mined a single-cell sequencing dataset of HCC and revealed that TAMs predominantly express GLUT3 but not GLUT1 (*Figure 3—figure supplement 4B*). Consistently, western blotting analysis showed a higher expression of GLUT3 and minimal levels of GLUT1 in THP-1 cells (*Figure 3—figure supplement 4C*). GLUT1 knockdown in THP-1 cells did not significantly affect their glycolytic metabolism, whereas GLUT3 knockdown led to a marked reduction in glycolysis in THP-1 cells (*Figure 3—figure supplement 4C, D*). Therefore, the effects of citalopram on macrophages might be primarily mediated through targeting C5aR1 rather than GLUT1.

In orthotopic GLUT1[KD] Hepa1-6 models, flow cytometry revealed a significant reduction of TAMs in the citalopram group compared with the control group (*Figure 3J*). Moreover, TAM polarization was shifted to a proinflammatory state, as evidenced by flow cytometry (*Figure 3K*) and real-time qPCR analysis (*Figure 3—figure supplement 5A*). Further phenotyping of tumor immune subsets in immunocompetent C57BL/6 mice showed that citalopram increased the infiltration of CD8[+] T and dendritic cells while decreasing the infiltration of CD4[+] T and B220[+] B cells (*Figure 3L*, *Figure 3—figure supplement 5B, C*). No significant changes were observed in splenic immune subsets upon citalopram treatment (*Figure 3—figure supplement 5D*).

Next, we sought to understand whether C5aR1[+] TAMs impact CD8[+] T cells. Intratumoral CD8[+] T cells from tumor-bearing *C5ar1*[−/−] mice had increased frequency and expression of GZMB, IFN-γ, and TNF-α compared with that from *C5ar1*[+/−] mice (*Figure 3M*). Likewise, the frequency and cytotoxicity of CD8[+] T cells from recipient *C5ar1*[−/−]; donor *C5ar1*[+/−] mice were reduced compared with recipient *C5ar1*[−/−]; donor *C5ar1*[−/−] mice, and this effect was partially reversed by citalopram treatment (*Figure 3N*). Furthermore, depletion of CD8[+] T cells abrogated the C5aR1[+] TAM-mediated enhancement of tumor growth (*Figure 3O*), suggesting that the anti-tumor effects of CD8[+] T cells are required for the pro-tumor phenotype of C5aR1[+] TAMs. Clinically, data from TCGA cohort reinforced the link between C5aR1 expression and an immunosuppressive TME as demonstrated by the associations with TAMs, exhausted T cells, immune checkpoint molecules, and effector Tregs (*Figure 3P*).

## Citalopram and CD8+ T cell activation

Considering the fact that citalopram targets GLUT1 and CD8+ T cell function is reliant on glycolytic metabolism (*Cao et al., 2023*; *Dong et al., 2024*), we investigated the in vivo effects of citalopram on CD8+ T cells. Citalopram significantly invigorated CD8+ T cell function and glycolytic metabolism in intrahepatic tumors from WT C57BL/6 mice (*Figure 4A, B*). To simulate the role of metabolic-associated fatty liver disease in the development of cirrhosis and HCC (*Du et al., 2022*; *Eslam et al., 2020*), we fed C57BL/6 mice a choline-deficient, amino acid-defined high-fat diet (CDAHFD) to induce metabolic dysfunction-associated steatohepatitis (MASH)/fibrosis prior to orthotopic Hepa1-6 cell implantation. Likewise, CD8+ T cell function and glycolytic metabolism were enhanced by citalopram treatment in MASH mice (*Figure 4C, D*). GLUT1-deficient cytotoxic T cells remain phenotypically and functionally unaffected (*Macintyre et al., 2014*), indicating that other glucose uptake mechanisms exist in cytotoxic T cells. GLUT1 and GLUT3 are essential for the activation and effector function of T cells (*Hochrein et al., 2022*; *Macintyre et al., 2014*). Similar to TAMs, intratumoral CD8+ T cells massively upregulated GLUT3 but not GLUT1 (*Figure 3—figure supplement 5E*), and this finding was supported by single-cell sequencing analysis (*Figure 3—figure supplement 4B*), suggesting the GLUT3-dependent glucose uptake in CD8+ T cells.

Given the wide range of peripheral 5-HT actions (*Jiang et al., 2021*), we hypothesized that citalopram modulates the immune system through the serotonergic mechanisms. Indeed, citalopram administered for 2 weeks in tumor-bearing mice led to reduced serum 5-HT levels (*Figure 4E*), which were accompanied by a reduction of low-grade systemic inflammatory responses in the MASH mice (*Figure 4F*). Compared to the control C57BL/6 mice, tumor-bearing *Tph1*−/− mice, which lack peripheral 5-HT, exhibited slowed tumor growth and increased effector function of intratumoral CD8+ T cells (*Figure 4G–I*). Interestingly, citalopram reduced serum 5-HT to a level as observed in *Tph1*−/− mice and displayed a superior inhibitory effect on tumor growth and CD8+ T cell function compared with that induced by *Tph1* deficiency (*Figure 4G–I*). These data not only corroborate recent reports that SSRIs modulate CD8+ T cell function via serotonergic-dependent mechanism (*Li et al., 2025*), but also reveal additional in vivo regulatory avenues by which citalopram affects CD8+ T cells, such as its ability to reprogram C5aR1+ TAMs. Notably, in the context of macrophage depletion, CD8+ T cell cytotoxicity was not further enhanced by citalopram (*Figure 3—figure supplement 1E*), indicating that TAM-dependent pathways are central to CD8+ T cell-mediated anti-tumor immunity and largely underlie the anti-tumor effects of citalopram.

Next, we investigated the direct effects of citalopram on T cell activation, expansion, and cytotoxicity. Citalopram enhanced αCD3/αCD28-stimulated CD8+ T cell proliferation as indicated by carboxy fluorescein succinimidyl ester (CFSE) staining (*Figure 4—figure supplement 1A*). Moreover, citalopram promoted the TCR-instigated activation of CD8+ T cells and the generation of effector CD8+ T cells as evidenced by the cell surface expression of CD44 and CD62L (*Figure 4—figure supplement 1B*). Likewise, the cytotoxic activity of activated CD8+ T cells was enhanced by citalopram, as reflected by GZMB, IFNγ, and TNF-α staining (*Figure 4—figure supplement 1C–E*).

Finally, to examine the immune-dependent anti-tumor effects of citalopram, we depleted either CD8+ or CD4+ T cells from GLUT1KD Hepa1-6 tumor-bearing mice and found that depletion of CD8+ T cells, rather than CD4+ T cells, increased tumor growth in citalopram-treated mice but had marginal effects in control mice (*Figure 4J*), showing that CD8+ T cells are essential for the effector mechanisms of citalopram.

## Discussion

Repurposing approved non-oncology drugs for cancer therapy offers cost-effectiveness, an accelerated development timeline, and a known safety profile as key advantages. Accumulating studies have deciphered the inverse link between SSRI use and cancer risk. However, there is a scarcity of studies investigating the potential repurposing SSRIs for cancer treatment and their mechanisms of action. Non-oncology drugs may have mechanisms of action that are different from traditional cancer treatments. Previously, we and others have uncovered several new targets of SSRIs, including GLUT1 in HCC (*Dong et al., 2024*), SMPD1 in glioblastoma (*Bi et al., 2021*), and VDAC1 in autophagy (*Hwang et al., 2021*). Here, we further broaden the knowledge of how SSRIs impact the immune system to exert anti-tumor effects. By targeting C5aR1 on TAMs, citalopram enhances the phagocytic activity

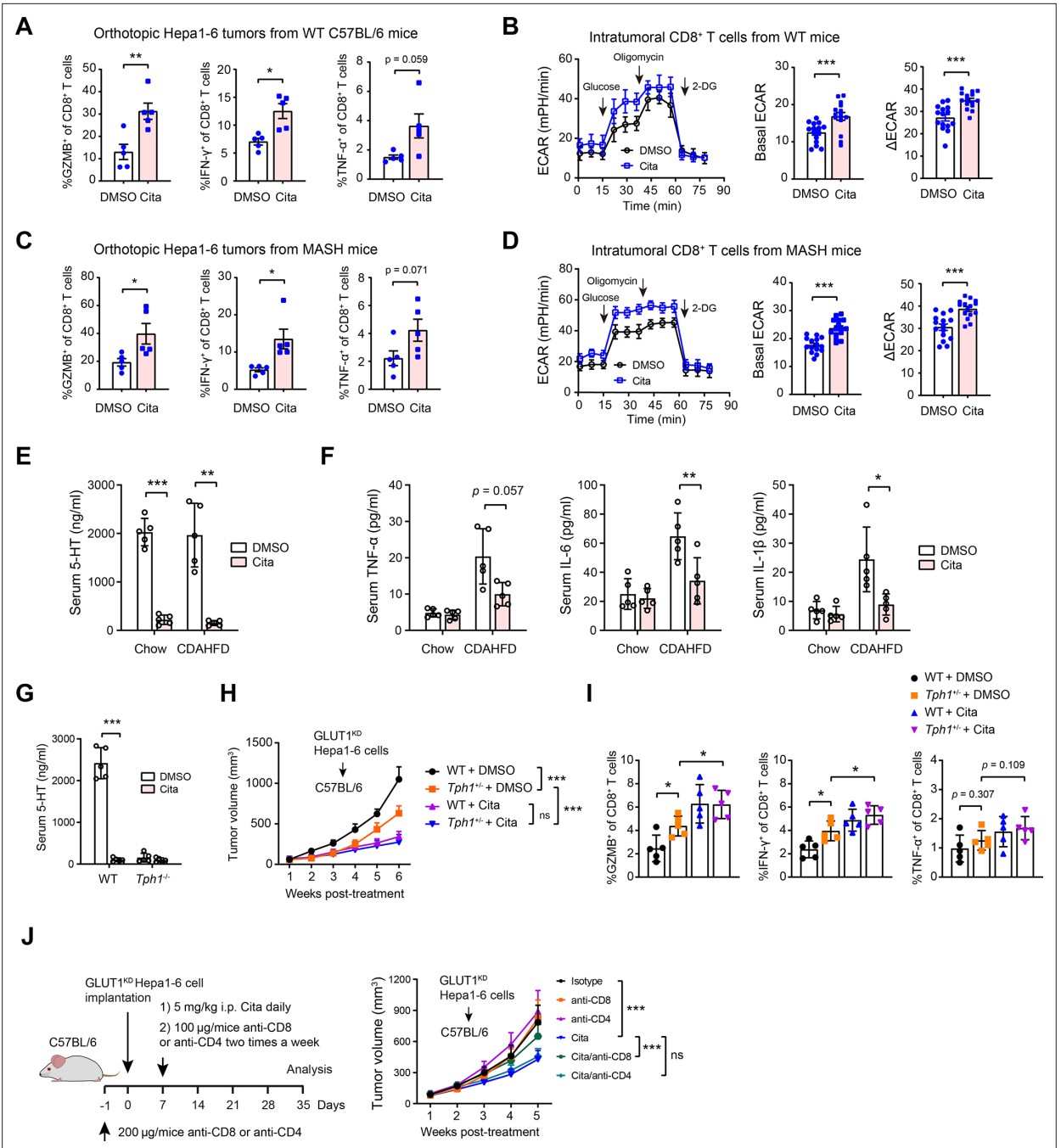

**Figure 4.** Citalopram activates CD8+ T cells. (**A, B**) Measurement of CD8+ T cell function and glycolysis in orthotopic tumor tissues from WT C57BL/6 mice (*n* = 5 per group). (**C, D**) Measurement of CD8+ T cell function and glycolysis in orthotopic tumor tissues from metabolic dysfunction-associated steatohepatitis (MASH) mice (*n* = 5 per group); basal extracellular acidification rate (ECAR) indicates glycolysis after the addition of glucose, and ΔECAR represents the difference between oligomycin-induced ECAR and 2-DG-induced ECAR. (**E**) Serum 5-hydroxytryptamine (5-HT) levels in GLUT1^KD Hepa1-6 tumor-bearing mice fed with chow diet or choline-deficient, amino acid-defined high-fat diet (CDAHFD), with the presence or absence of citalopram treatment (*n* = 5 per group). (**F**) Serum TNF-α, IL-1β, and IL-6 levels in GLUT1^KD Hepa1-6 tumor-bearing mice fed with chow diet or CDAHFD, with the presence or absence of citalopram treatment (*n* = 5 per group). (**G**) Serum 5-HT levels in WT C57BL/6 and *Tph1*^−/− mice, with the presence or absence of citalopram treatment (*n* = 5 per group). (**H**) Tumor growth of WT and *Tph1*^−/− mice after subcutaneous injection of Hepa1-6 cells and treatment with citalopram. (**I**) Measurement of CD8+ T cell function in tumor tissues from the groups mentioned in **H**. (**J**) The therapeutic effect of citalopram on GLUT1^KD Hepa1-6 tumor was tested in the presence or absence of CD4+ T or CD8+ T cell depletion. In all panels, *p < 0.05, **p < 0.01, ***p < 0.001; ns, non-significant. Values are presented as mean ± SD and compared by the Student's *t* test (**A–G**), one-way analysis of variance (ANOVA) multiple comparisons with Tukey's method (**I**), and two-way ANOVA with Dunnett's multiple comparisons (**H, J**).

*Figure 4 continued on next page*

*Figure 4 continued*

The online version of this article includes the following figure supplement(s) for figure 4:

**Figure supplement 1.** The in vitro effects of citalopram treatment on CD8[+] T cell function.

of TAMs and boosts local anti-tumor immune responses, particularly CD8[+] T cell anti-tumor immunity. Furthermore, citalopram has a direct stimulatory effect on the activation of CD8[+] T cells in vitro (*Figure 5*).

C5aR1 is expressed on TAMs within the TME, which helps to explain why citalopram has better anti-tumor effects in immune-competent mice. In addition to TAMs, C5aR1 is also expressed by neutrophils, mast cells, and myeloid-derived suppressor cells in cancers (*Ding et al., 2020*; *Medler et al., 2018*; *Ou et al., 2021*). Although the functional impacts of citalopram on these C5aR1-expressing immune cells have not been formally determined, macrophage depletion assay and *C5ar1⁻/⁻* BM transfer experiment suggested that citalopram exerted anti-tumor functions in vivo through C5aR1-expressing TAMs. Targeting C5aR1 can reprogram TAMs from a pro-tumor state to an anti-tumor state, promoting the secretion of CXCL9 and CXCL10 while facilitating the recruitment of cytotoxic CD8[+] T cells (*Luan et al., 2024*; *Medler et al., 2018*). In alignment with this, citalopram treatment increased CD8[+] T cell filtration and their cytotoxic activities in HCC, and this effect was largely dependent on C5aR1[+] macrophages. Citalopram induces TAM phenotypic polarization toward an M1 proinflammatory state (*Zhao et al., 2024*), which supports anti-tumor immune response within the TME. Another possibility is that citalopram targets C5aR1 to enhance macrophage phagocytosis and subsequent antigen presentation and/or cytokine production, which promotes T cell recruitment and activity

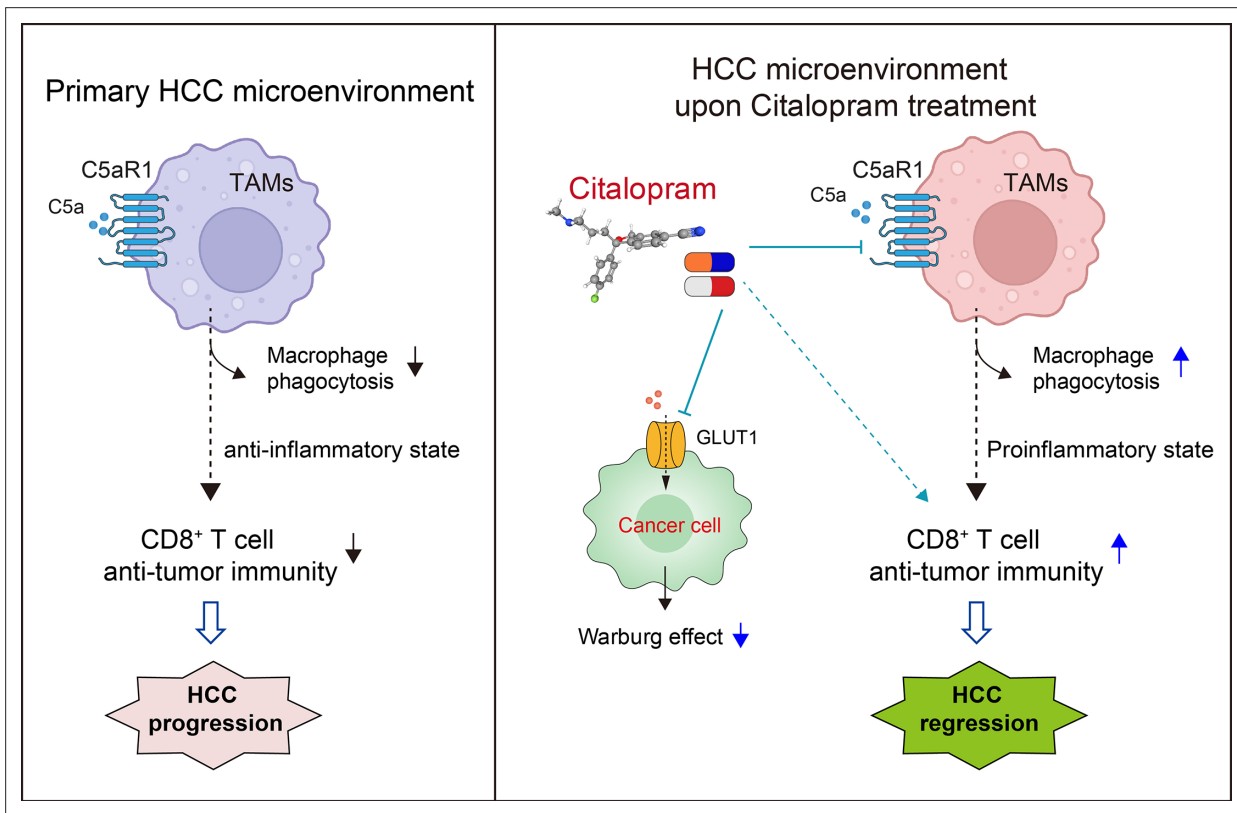

**Figure 5.** Mechanism model. Model depicting the molecular mechanism by which citalopram inhibits the Warburg effect and promotes an anti-tumor response in hepatocellular carcinoma (HCC). In the primary HCC microenvironment (left panel), C5aR1-expressing tumor-associated macrophages (TAMs) exhibit reduced phagocytic capacity and an anti-inflammatory state, which correlates with diminished CD8[+] T cell anti-tumor immunity and HCC progression. Upon treatment with citalopram (right panel), the drug not only inhibits the glycolytic metabolism of cancer cells by targeting GLUT1 (*Dong et al., 2024*) but also acts on C5aR1 expressed by TAMs, thereby enhancing macrophage-driven anti-tumor immunity. Additionally, citalopram induces a systemic immunostimulatory effect on CD8[+] T cell functions through yet-to-be-identified serotonergic mechanisms. The dotted line indicates a causal relationship that has not been fully established through direct evidence.

as well as modulates other aspects of tumor immunity. Therefore, we provide a mechanistic insight regarding how citalopram targets C5aR1 activity in TAMs. The central findings are important to our understanding of citalopram-induced anti-tumor activities and may lay a basis for upcoming mechanistic studies aimed at deciphering specific tumor-promoting roles of this defined TAM population.

While we highlight the anti-HCC effects of SSRIs through targeting C5aR1, it is possible that SSRIs impact tumor progression by modulating the systemic environment via serotonergic mechanisms, especially concerning the activation of CD8+ T cells. Apart from its role in regulating hemostasis, 5-HT exerts immunoregulatory functions on circulating immune cells (*Costa et al., 2020*). In this study, it was observed that both citalopram treatment and *Tph1* deficiency enhanced CD8+ T cell cytotoxic activity. Peripheral 5-HT has been shown to enhance PD-L1 expression on cancer cells, thereby impairing functions of CD8+ T cells within tumors (*Schneider et al., 2021*). 5-HT deficiency delays the growth of syngeneic pancreatic and colorectal tumors by promoting the accumulation and effector functions of CD8+ T cells (*Schneider et al., 2021*). In addition to the traditional receptor-dependent signaling transduction, 5-HT can modulate CD8+ T activity through post-translational modification known as serotonylation (*Wang et al., 2024*). CD8+ T cells express TPH1 (*Li et al., 2025*; *Wang et al., 2024*), enabling them to synthesize endogenous 5-HT, which activates their activity through serotonylation-dependent mechanisms. It remains to be explored whether serotonylation contributes to the immunomodulatory effects of citalopram, as citalopram directly stimulates CD8+ T cell proliferation, activation, and cytotoxicity in vitro. The observed reduction in serum 5-HT levels and activation of CD8+ T cells following SSRI treatment further highlight the complex interplay between neurotransmitters, the immune system, and cancer progression. Although citalopram directly stimulates CD8+ T cells in vitro, the TAM-independent activation is not evident in vivo within the complex TME, as CD8+ T cell responses are abolished by macrophage depletion, indicating that the in vivo effects of citalopram on CD8+ T cells and tumor growth are largely TAM-dependent.

Citalopram docks into the GLUT1 pocket with an electrostatic contact to E380 and surrounding hydrophobic contacts (*Dong et al., 2024*). Although GLUT1 and GLUT3 share a highly conserved core substrate-binding pocket, isoform-specific regulation arises from features outside the canonical site. Structural homology between GLUT1 and GLUT3 is high in the transmembrane core, but regulatory features, such as the cytosolic Sugar Porter (SP) motif network, the conserved A motif, lipid interfaces, and gating dynamics, differ between the two isoforms (*Custódio et al., 2021*). These differences can alter pocket accessibility, coupling to conformational transitions, and allosteric signaling to the cytosol, such that a GLUT1 inward-facing binder may not stabilize a GLUT3 conformation that yields appreciable transport inhibition. Consistently, functional experiments show robust GLUT1 engagement in cancer cells (*Dong et al., 2024*), while equivalent GLUT3 inhibition has not been observed in TAMs (*Figure 3—figure supplement 4*), supporting isoform-selective targeting by citalopram in this context.

In a study involving 308,938 participants with HCC, findings indicated that the use of antidepressants following an HCC diagnosis was linked to a decreased risk of both overall mortality and cancer-specific mortality (*Huang et al., 2023*). These associations were consistently observed across various subgroups, including different classes of antidepressants and patients with comorbidities such as hepatitis B or C infections, liver cirrhosis, and alcohol use disorders (*Huang et al., 2023*). Similarly, our analysis of real-world data from the Swedish Cancer Register demonstrated that SSRIs are correlated with slower disease progression in HCC patients (*Dong et al., 2024*). Given these insights, antidepressants, especially SSRIs, show significant potential as anticancer therapies for individuals diagnosed with HCC. Immunotherapy, especially immune checkpoint inhibitors (ICIs), has revolutionized the management of HCC (*Zheng et al., 2021*). The immune microenvironments play crucial roles in the response or resistance of HCC to immunotherapies (*Llovet et al., 2022*). However, limited molecular biomarkers are available to guide clinical decision-making for ICI therapy of HCC. Given SSRIs can increase CD8+ T cell infiltration as well as cytotoxic activities by targeting C5aR1+ TAMs, one can suppose that C5aR1+ TAMs within the HCC microenvironment might be used as a molecular marker for SSRIs use in combination with ICI therapy.

There are several limitations in this study. First, the potential promiscuity of citalopram's interactions across GLUT1, C5aR1, and SERT1, as varying affinities could influence the drug's overall efficacy. Citalopram interacts with C5aR1 and GLUT1 through distinct binding sites and mechanisms, whereas its interaction with SERT is characterized by a more direct inhibition of serotonin binding (*Coleman*

*et al., 2016*). Employing techniques such as surface plasmon resonance or biolayer interferometry could provide valuable quantitative data on binding kinetics and affinities for each target. Second, it is challenging to dissect how citalopram's interactions with multiple targets may contribute to its therapeutic effects, particularly in the context of immune modulation and tumor progression. The potential for citalopram to exhibit diverse mechanisms of action through these interactions warrants further investigation.

In conclusion, our study suggests the potential repurposing of SSRIs for the treatment of HCC and highlights C5aR1 as a direct target of citalopram. Citalopram exerts its effects through an immune-mediated pathway, primarily by targeting C5aR1$^+$ TAMs. Combining SSRIs with ICIs or other first-line therapies could enhance the treatment efficacy of current standard therapies.

## Materials and methods

### Animal models

In the subcutaneous xenograft model, *Rag1*$^{-/-}$ C57BL/6 mice (male, 6- to 8-week-old, Cyagen Biosciences, C001197), wild type immune-competent C57BL/6 mice (male or female, 6- to 8-week-old) or *Tph1*$^{-/-}$ mice (male, 6- to 8-week-old, Cyagen Biosciences, S-KO-05541), were maintained on a 12-hr day/night cycle with ad libitum access to food and water. For the pharmacological inhibition study, 5 × 10$^6$ mouse HCC cells (wild type, shControl, sh*Slc6a4*, or sh*Slc2a1* Hepa1-6 and Hep53.4) in 100 μl phosphate-buffered saline (PBS) were injected subcutaneously in the lower back. When bore visible tumors, mice were given intraperitoneal injections of vehicle or citalopram (5 mg/kg body weight, Selleck, Shanghai, China) daily (*n* = 5–7 per group).

In the orthotopic xenograft model, 5 × 10$^6$ Hepa1-6 cells transfected with luciferase-expressing lentiviruses were orthotopically implanted into the liver of WT or MASH C57BL/6 mice. The day following tumor cell implantation, mice were treated with citalopram (5 mg/kg body weight) or DMSO. In the diet-induced MASH model, mice were fed a CDAHFD (45 kcal% fat) supplemented with 0.1% methionine (CDAHFD diet, A06071309, from Research Diets) for a duration of 6 weeks.

For the C5aR1 knockdown study, C57BL/6J-*C5ar1*$^{em1Cya}$ mice were purchased from Cyagen Biosciences (S-KO-01274, Suzhou, China). 5 × 10$^6$ sh*Slc2a1* Hepa1-6 cells were injected into the lower back of 6- to 8-week-old male *C5ar1*$^{+/-}$ or *C5ar1*$^{-/-}$ C57BL/6 mice. Tumor volume was measured every week using a digital caliper. Four weeks later, mice were sacrificed and xenograft tumors were analyzed.

To determine the association between citalopram and C5aR1, BM transfer experiments were performed. Six- to eight-week-old male *C5ar1*$^{-/-}$ C57BL/6 mice were served as recipients. Mice underwent full-body irradiation for conditioning. BM was harvested from the femurs and tibias of donor mice carrying C5ar1 genotypes *C5ar1*$^{+/-}$ or *C5ar1*$^{-/-}$, and 1.0 × 10$^7$ cells were delivered intravenously into each recipient. After transplantation, mice were given antibiotic-supplemented water for 4 weeks to support engraftment. Ten weeks post-transplant, the recipients were considered fully immune-reconstituted and were used for GLUT1$^{KO}$ Hepa1-6 tumor challenge experiments. A total of 2 × 10$^6$ GLUT1$^{KO}$ Hepa1-6 cells were injected subcutaneously into the lower back of each recipient. One week after cancer cell implantation, mice were subjected to treatment with citalopram (5 mg/kg body weight) or DMSO. At indicated time points, mice were sacrificed and xenograft tumors were analyzed.

In all animal studies, tumor volume was assessed using caliper measurements and calculated using the formula: tumor = 1/2(length × width$^2$), where length represents the largest diameter and width indicates the smallest diameter. At the experimental endpoint, mice were humanely euthanized, tumors were excised, and tumor weight was recorded. All animals received proper care in accordance with the guidelines outlined in the 'Guide for the Care and Use of Laboratory Animals' established by the National Academy of Sciences and published by the National Institutes of Health (NIH publication 86-23 revised 1985). The protocols for animal studies were approved by the ethics committee of the Ren Ji Hospital, School of Medicine, Shanghai Jiao Tong University (approval number 202201427 and RA-2021-096).

### Cell culture and reagents

The following cell lines were included: Hepa1-6, Hep53.4, THP-1, and HEK293T. All cells were cultured in a medium recommended by ATCC, supplemented with 10% fetal bovine serum (FBS; Gibco, USA) and 1% (vol/vol) streptomycin–penicillin (Sigma-Aldrich, Shanghai, China), and maintained at 37°C in a

humidified atmosphere with 5% $CO_2$. Prior to conducting cell experiments, all the mentioned cell lines underwent testing for short tandem repeat profiling and mycoplasma contamination to ensure their authenticity and purity. Citalopram (S4749), escitalopram (S4064), fluoxetine (S1333), fluvoxamine (S1336), paroxetine (S3005), and sertraline (S4053) were all obtained from Selleck (Shanghai, China).

## Plate colony formation assay

For cell proliferation analysis, a total of 1,000 indicated HCC cells were seeded into 6-well plates with three replicates. After treatment with citalopram for 10–14 days, the plates were washed with PBS twice, fixed with methanol for 10 min, and stained with 0.5% (wt/vol) crystal violet solution for 10 min. Colonies containing more than 30 cells were counted under the microscope.

## Detection of caspase-3/7 activity

Cell apoptosis in HCC cells upon citalopram treatment was assessed using the Apo-ONE Homogeneous Caspase-3/7 Assay (Promega, Madison, WI, USA) as described previously (*Jiang et al., 2017*). To normalize the data, the cell viability was detected using CellTiter-Blue (Promega, G8081). The relative caspase-3/7 activity was determined by calculating the ratio of the signals obtained from the Apo-ONE assay to those from the CellTiter-Blue assay.

## Gene knockdown and knockout

The specific shRNA oligonucleotides targeting *Slc6a4* and *Slc2a1* were custom-synthesized by Genepharma (Shanghai, China). Lentivirus packaging was carried out in HEK293T cells using Lipofectamine 2000 (Invitrogen, Carlsbad, CA), and virus titers were determined following standard procedures. HCC cells (Hep53.4 and Hepa1-6) were infected with the virus-containing supernatant in the presence of 6 µg/ml polybrene (Sigma-Aldrich, H9268, St. Louis, MO) when they reached 70% confluence. Cells expressing the shRNAs were selected using puromycin (2 µg/ml, Gibco, A1113802).

For CRISPR/Cas9-mediated gene knockout, THP-1 cells were initially transfected with the lentiCas9-Blast vector to facilitate the expression of Cas9. To generate *C5AR1*$^{-/-}$ THP-1 cells, a lentiviral system utilizing guide RNA (sgRNA) targeting *C5AR1* was utilized. The lentivirus was produced in HEK293T cells by transfecting lentiGuide-Puro along with packaging plasmids pVSVg and psPAX2. The transfection was carried out using Fugene6 (E2693, Promega) in OptiMEM. After 2 days, the supernatants were filtered through a 0.45-µm filter, and Cas9-expressing cells were infected with the lentivirus in the presence of 6 µg/ml polybrene. Individual colonies were then seeded into 96-well plates, allowed to reach confluence, and subsequently subjected to knockout efficiency analysis. The sequences for shRNA and sgRNA are provided below: sh*Slc6a4*, CACCGCCTCCTACTATAACACCATCCGAAGAT GGTGTTATAGTAGGAGGC; sh*Slc2a1*, CACCGGGAGAAGAAGGTCACCATCTCGAAAGATGGT GACCTTCTTCTCCC; *C5AR1* sgRNA, CTTCAGTCAACACGTTCCGG.

## Quantitative real-time PCR

RNA was extracted from CD8$^+$ T cells using the RNAiso Plus reagent (Takara, Japan) and subjected to reverse transcription using the PrimeScript RT-PCR kit (Takara, Japan). Then, quantitative real-time PCR was performed using the 7500 Real-time PCR system (Applied Biosystems, USA). The housekeeping gene *Gapdh* was used to normalize gene expression. Primer sequences used in this study were shown below: *Slc2a1* forward, 5'-CAGTTCGGCTATAACACTGGTG-3'; *Slc2a1* reverse, 5'-GCCC CCGACAGAGAAGATG-3'; *Slc2a3* forward, 5'-ATGGGGACAACGAAGGTGAC-3'; *Slc2a3* reverse, 5'-GTCTCAGGTGCATTGATGACTC-3'; *Gapdh* forward, 5'-AGGTCGGTGTGAACGGATTTG-3'; *Gapdh* reverse, 5'-TGTAGACCATGTAGTTGAGGTCA-3'.

## Immunoblotting

Cellular lysates were extracted using RIPA lysis buffer (P0013B, Beyotime, Shanghai, China) supplemented with protease and phosphatase inhibitor cocktails (ab201119, Abcam, Shanghai, China). The protein concentration was quantified with a BCA Protein Assay Kit (Pierce Biotechnology, USA). The proteins were then separated by sodium dodecyl sulfate–polyacrylamide gel electrophoresis and transferred onto polyvinylidene difluoride (Millipore) membranes. Subsequently, the membranes were blocked with 5% (m/v) skim milk for 1 hr at room temperature before being incubated with specific antibodies overnight at 4°C. On the following day, the membranes were

washed three times with PBS and then incubated with HRP-conjugated secondary antibodies for 45 min. Subsequently, an ECL chemiluminescence assay (SB-WB012, Share-bio, Shanghai, China) was performed using a Bio-Spectrum Gel Imaging System (Bio-Rad). The antibodies utilized in this study included: SERT (1:500, ProteinTech, 19559-1-AP), GLUT1 (1:2,000, Cell Signaling Technology, #73015), GLUT3 (1:1,000, ProteinTech, 20403-1-AP), C5aR1 (1:1,500, ProteinTech, 21316-1-AP), and β-actin (1:2,000, Abcam, ab8226). The information for secondary antibodies is as follows: goat anti-rabbit IgG (H+L) cross-adsorbed secondary antibody (1:5,000, Thermo Fisher Scientific, G-21234) and goat anti-mouse IgG (H+L) cross-adsorbed secondary antibody (1:5000, Thermo Fisher Scientific, G-21040).

## Immunohistochemistry and immunofluorescence analysis

Paraffin-embedded sections of mouse HCC tissues were subjected to immunohistochemical analysis of Ki67, CCS3, and C5aR1 expression. Immunohistochemical analysis was performed as reported previously (*Jiang et al., 2017*). The primary antibodies used in this study were shown as follows: Ki67 (1:400, Cell Signaling Technology, #9449), CCS3 (1:400, Cell Signaling Technology, #9661), and C5aR1 (1:1,200, ProteinTech, 21316-1-AP). For immunofluorescence analysis, paraffin sections (5 μm) of HCC samples were deparaffinized and rehydrated with graded ethanol, followed by microwave heating-based antigen retrieval. Then tissue sections were blocked with 10% donkey serum (Sigma-Aldrich, D9663, USA) for 1 hr at room temperature and incubated with primary antibodies at 4°C overnight. The following antibodies were used: C5aR1 (1:50, Proteintech, 10375-1-AP) and CD163 (1:200, Abcam, ab182422). The next day, tissue sections were incubated with donkey anti-rabbit Alexa Fluor 594 (1:400, Jackson ImmunoResearch, #711-585-152) and donkey anti-mouse Alexa Fluor 488 (1:400, Jackson ImmunoResearch, #715-545-150) for 30 min at room temperature. Finally, 4′,6-diamidino-2-phenylindole staining was performed to visualize the nuclei. A confocal microscope equipped with a digital camera (Nikon, Japan) was used for high-resolution imaging.

## Bioinformatics analysis and single-cell sequencing data

For GSEA analyses, we conducted RNA sequencing (RNA-seq) analysis on HCC-LM3 cells treated with citalopram or fluvoxamine, which led to the identification of 114 differentially expressed genes (DEGs; 80 co-upregulated and 34 co-downregulated), as reported previously (*Dong et al., 2024*) (PRJNA1084911). These DEGs were then utilized to create an SSRI-related gene signature. Subsequently, we analyzed RNA-seq data from liver HCC (LIHC) samples in The Cancer Genome Atlas (TCGA) cohort, comprising 371 samples, categorizing them into high and low expression groups based on the median expression levels of each candidate target gene. Finally, we performed GSEA on the grouped samples using the SSRI-related gene signature. Correlation analysis was performed with GEPIA2 (http://gepia2.cancer-pku.cn/#index; *Tang et al., 2019*). The following gene signatures were included for analysis: TAMs (*CD33, MARCO, CXCL5, SULT1C2, MSR1, CTSK, PTGDS, COLEC12, GPC4, PCOLCE2, CHIT1, CLEC5A, CCL7, FN1, CD163, GM2A, BCAT1, RAI14, COL8A2, CHI3L1, ATG7, CD84, MS4A4A, EMP1, CYBB, CD68,* and *CD11b*), exhausted T cells (*HAVCR2, TIGIT, LAG3, PDCD1, CXCL13,* and *LAYN*), immune checkpoint (*IDO1, IDO2, PDCD1, CD274, PDCD1LG2, CTLA4, LAG3, HAVCR2, C10orf54, BTLA, ICOS,* and *TNFRSF9*), and effector Treg cells (*FOXP3, CTLA4, CCR8,* and *TNFRSF9*). The single-cell transcriptome expression of C5aR1 and GLUT family members in the TME was identified with GSE125449 and GSE140228-10X.

## DARTS experiments

THP-1 or HEK293T cells were lysed using NP40 lysis buffer (Thermo Fisher Scientific, 89842Y). Following centrifugation at 12,000 rpm for 10 min, the supernatant containing the proteins was collected, and the protein concentration was determined using a BCA Protein Assay Kit (Pierce Biotechnology, USA). Samples of equivalent quality were then treated with specific SSRIs and a vehicle at 37°C for 4 hr. Subsequently, pronase (10 μg/ml, Roche, 10165921001) or distilled water was added and incubated at room temperature for specified time points (10 and 30 min). The reaction was terminated by adding protease inhibitors, and the resulting products were harvested and subjected to analysis using western blotting.

## In silico docking study

To predict the binding mode of citalopram/C5aR1, molecular docking was performed using the AutoDock 4.2.6 software package (*Morris et al., 2009*). The receptor structure for C5aR1 (PDB id: 6C1Q) (*Liu et al., 2018*) was sourced from the RCSB protein data bank (https://www.rcsb.org/). Due to the lack of available structural information, the structure of mouse C5aR1 was predicted using a local version of the ColabFold (AlphaFold2) software (*Jumper et al., 2021*; *Mirdita et al., 2022*). The original substrates and other heteroatoms in these crystal structures were removed for further docking procedure. The absolute structure of citalopram (CID: 2771), fluoxetine (CID: 3386), fluvoxamine (CID: 5324346), paroxetine (CID: 43815), and sertraline (CID: 68617) was sourced from the PubChem database (https://pubchem.ncbi.nlm.nih.gov/). In each docking scenario, molecular docking was conducted following a standard protocol (*Forli et al., 2016*), with a modification in the number of genetic algorithm runs to 500 to comprehensively explore the binding modes. The binding energy and cluster information were directly extracted from the AutoDock output.

## Biochemical analysis

The serum levels of inflammatory cytokines (TNF-α, IL-1β, and IL-6) were analyzed using commercial ELISA kits following the manufacturer's instructions (R&D Systems). The cytokine concentrations were calculated using a standard curve, and the data were presented in picograms per milliliter. Serum 5-HT levels were assessed using the 5-HT ELISA Kit from Abcam (ab133053), following the manufacturer's guidelines.

## Isolation of TAMs from xenograft tumor tissues

Fresh sterile tumor tissues from subcutaneous xenograft model were washed with PBS two times, minced with scalpels, and then digested using a Tumor Dissociation Kit (Miltenyi Biotec, Cat #130-095-929) at room temperature for 30 min. Digested tumor tissues were filtered through Falcon 70Micrometer Cell Strainer (Corning, #352350) and centrifuged at 1500 rpm for 5 min. Red blood cells were removed using Red Blood Cell Lysis Solution (Miltenyi Biotec, Cat #130-094-183). Then, TAMs were selected by CD11b microbeads (Miltenyi Biotec, Cat #130-049- 601) according to the manufacturer's instructions.

## Isolation of immune cells and flow cytometry

Mice were euthanized, and fresh tumor tissues and spleen were harvested. Spleens were mechanically disrupted through 70 µm Nylon mesh (Corning, #352350) to release single cells. The cells were washed with PBS, followed by the lysis of red blood cells with Red Blood Cell Lysis Solution (Miltenyi Biotec, Cat #130-094-183). Tumor tissues were cut into small pieces with surgical scissors and digested with 2 mg/ml collagenase IV (Sigma) for 20 min and filtered with 70 µm Nylon mesh. The isolated cells were collected with PBS and subjected to centrifugation at 1500 rpm for 5 min. To enrich immunocytes from tumor tissues, harvested cells were centrifuged through different concentrations of discontinuous gradients of 40/80 Percoll (GE Healthcare, Cat #17-0891-01) at 350 × *g* for 30 min. The middle pellet was washed with PBS, subjected to red blood cell lysis for 3 min.

For staining molecules expressed on cell surface, single-cell suspensions from spleens and tumor tissues were labeled with fluorophore-conjugated antibodies: PerCP-Cyanine 5.5 7AAD (Biolegend, Cat #420404, 1:1000), FITC anti-mouse CD4 (Biolegend, Cat #100406, 1:1000), PE anti-mouse Ly6G (Biolegend, Cat #127608, 1:200), APC anti-mouse CD11c antibody (Biolegend, Cat #117310, 1:200), PE-cy7 anti-mouse CD8α (Biolegend, Cat #100722, 1:200), Brilliant Violet 421 anti-mouse CD3 (Biolegend, Cat #100227, 1:200), Alex Fluor 647 anti-mouse F4/80 (Invitrogen, Cat #2604384, 1:200), Brilliant Violet 510 anti-mouse CD45 (Biolegend, Cat #103138, 1:200), PE-cy7 anti-mouse NK1.1 (Biolegend, Cat #156513, 1:200), Brilliant Violet 421 anti-mouse B220 (Biolegend, Cat #103240, 1:200), Brilliant Violet 605 anti-mouse CD11b (Biolegend, Cat #101257, 1:200), and APC-CY7 Stain 780 (BD Bioscience, Cat #565388, 1:1000).

For intracellular staining, single-cell suspensions from tumor tissues were stimulated with Cell Stimulation Cocktail (Invitrogen) for 4 hr and labeled with PE anti-mouse CD45 antibody (Biolegend, Cat #103106, 1:200), PerCP-Cyanine 5.5 CD3 (Biolegend, Cat #100325, 1:200), PE-cy7 anti-mouse CD8a (Biolegend, Cat #100722, 1:200), APC-CY7 Stain 780 (BD Bioscience, Cat #565388, 1:1000). After fixation and permeabilization using the Fix/Perm kit (Thermo Fisher, GAS004), cells were labeled with

APC anti-human/mouse Granzyme B Recombinant antibody (Biolegend, Catalog No. 372203, 1:200), Brilliant Violet 421 anti-mouse IFN-γ (Biolegend, Cat #505830, 1:200), and Brilliant Violet 605 anti-mouse TNF-α (Biolegend, Cat #506329, 1:200). Flow cytometry was performed in BD FACSCanto II.

## T cell activation, expansion, and cytotoxicity

Single-cell suspensions were freshly dissected from the spleen and lymph nodes, and red blood cells were removed using Red Blood Cell Lysis Solution (Miltenyi Biotec, Cat #130-094-183). CD8$^+$ T cells were isolated from the resulting cell suspensions by CD8$^+$ magnetic bead selection (Miltenyi Biotec, Cat #130-117-044). For T cell proliferation assay, isolated CD8$^+$ T cells were resuspended in PBS supplemented with CFSE (Invitrogen, 1:10,000) for 8 min at room temperature and washed with RPMI 1640, supplemented with 10% heat-inactivated FBS. Then, CD8$^+$ T cells were activated by plate-bound αCD3ε (5 µg/ml, Biolegend, Catalog No. 100359) and α-CD28 (2 µg/ml, Biolegend, Catalog No. 102121) in complete RPMI 1640 medium at 37°C and 5% CO$_2$ for 72 hr in the presence or absence of 5 µM citalopram treatment. Finally, cell proliferation was assessed by flow cytometry. For assessment of cellular activation and cytotoxicity, CD8$^+$ T cells were stimulated by Leukocyte Activation Cocktail (GolgiPlug BD, Catalog no.550583) for 4 hr, resuspended in PBS/2% FBS containing PE anti-mouse CD62L antibody (Biolegend, Catalog No. 104407, 1:200) and Brilliant Violet 510 anti-mouse/human CD44 Antibody (Biolegend, Catalog No. 103044, 1:200), followed by incubation at room temperature for 30 min and washed in PBS/2% FBS. After fixation and permeabilization using the Cytofix/Cytoperm solution (BD Biosciences) for 20 min at 4°C, Brilliant Violet 421 anti-mouse IFN-γ antibody (Biolegend, Catalog No. 505830, 1:200), APC anti-human/mouse Granzyme B Recombinant antibody (Biolegend, Catalog No. 372203, 1:200), and Alexa Fluor 488 anti-mouse TNF-α antibody (Biolegend, Catalog No. 506313, 1:200) were employed for intracellular staining. Then, cell activation and cytotoxicity were assessed by flow cytometry.

## Macrophage phagocytosis assay

Phagocytosis assays of tumor cells were performed as previously described (*Chen et al., 2017*). Briefly, 5 × 10$^4$ phorbol myristate acetate (100 ng/ml) primed THP-1 cells and harvested TAMs were seeded overnight in a 24-well tissue culture plate. The next day, HCC-LM3 cells and Hepa1-6 cells were labeled with pHrodo Green (Thermo, Cat #P35373, 1:1000). After incubating macrophages in serum-free medium for 2 hr, the macrophages were labeled with CellTracker Deep Red Dye (Thermo, Cat #C34565, 6 ng/ml). Then, 5 × 10$^4$ predyed tumor cells were added to the macrophages, in the presence or absence of C5a stimulation and different SSRIs (5 µM). After incubation at 37°C for 4 hr, macrophages were extensively washed and imaged with Confocal Laser Scanning Microscope (Leica).

## Macrophage depletion assay

The method for eliminating macrophages was reported previously (*Jiang et al., 2019*). Briefly, C57BL/6 mice were intraperitoneally injected with clodronate liposomes (1.4 mg/20 g body weight) to remove macrophages. Clodronate liposomes (Yeasen, 40337ES10, Shanghai, China) were given every 3 days for 12 days before tumor inoculation. After inoculation of sh*Slc2a1* (GLUT1$^{KD}$) Hepa1-6 cells, clodronate liposomes were also administered every 3 days until the end of the experiment. When the mice were sacrificed, liver and tumor tissues were extracted, and F4/80 staining (1:250, Cell Signaling Technology, #70076) was performed to determine the efficiency of macrophage depletion.

## CD4$^+$ T and CD8$^+$ T cell depletion experiments

For CD4$^+$ T and CD8$^+$ T cell depletion, mice were treated intraperitoneally with 200 µg of specific monoclonal antibodies (Ultra-LEAF Purified anti-mouse CD8a Antibody, Biolegend, Cat #100763; and Ultra-LEAF Purified anti-mouse CD4 Antibody, Biolegend, Cat #100457) the day before the implantation of cancer cells and then with 100 µg twice a week for the entire duration of the experiment.

## Statistical analysis

The sample sizes for in vitro and in vivo studies were determined based on similar studies in the field and through pilot studies. Details regarding sample sizes and biological replicates can be found in the figure legends. Continuous variables are presented as means ± SD. For comparisons between two independent groups, a two-tailed Student's *t* test was employed. Statistical analyses involving

more than two groups were conducted using either a one-way analysis of variance (ANOVA) with Tukey's method or two-way ANOVA with Dunnett multiple comparisons, as specified. GraphPad Prism 9 (GraphPad Software Inc, San Diego, CA) was used for statistical analyses. Kaplan–Meier survival curves were generated and analyzed using the log-rank test. A p-value less than 0.05 was considered statistically significant. In all figures, statistical significance is represented as follows: *$p < 0.05$; **$p < 0.01$; ***$p < 0.001$.

## Acknowledgements

The research was supported by grants from National Natural Science Foundation of China (82472883), Shanghai Pilot Program for Basic Research-Shanghai Jiao Tong University (21TQ1400225), China Foundation For Youth Entrepreneurship and Employment (P24062487785), 'Rising Stars of Medical Talents' Youth Development Program (Youth Medical Talents-Specialist Program), Clinical Medicine Research Center Construction Project of Huadong Hospital (LCZX2202), Shanghai Outstanding Young Medical Personnel Training Program & Excellence Project of Shanghai Municipal Health Commission (20224Z0009), Key specialized diseases constructionSpecialized Diseases Construction of Huadong Hospital (ZDZB2225), Key Areas Research and Development Programs of Guangdong Province (2023B1111050009), and

Shanghai Oriental Talent Program – Youth Project (QNWS2024017).

## Additional information

### Funding

| Funder | Grant reference number | Author |
| --- | --- | --- |
| National Natural Science Foundation of China | 82472883 | Fangyuan Dong |
| Basic Research-Shanghai Jiao Tong University | 21TQ1400225 | Helen He Zhu |
| China Foundation for Youth Entrepreneurship and Employment | P24062487785 | Fangyuan Dong |
| Clinical Medicine Research Center Construction Project of Huadong Hospital | LCZX2202 | Zhijun Bao |
| Shanghai Outstanding Young Medical Personnel Training Program & Excellence Project of Shanghai Municipal Health Commission | 20224Z0009 | Xiaona Hu |
| Key Specialized Diseases Construction of Huadong Hospital | ZDZB2225 | Xiaona Hu |
| Key Areas Research and Development Programs of Guangdong Province | 2023B1111050009 | Zhi-Gang Zhang |
| Shanghai Oriental Talent Program – Youth Project | QNWS2024017 | Fangyuan Dong |

The funders had no role in study design, data collection, and interpretation, or the decision to submit the work for publication.

### Author contributions

Fangyuan Dong, Resources, Data curation, Formal analysis, Funding acquisition, Validation, Investigation, Methodology, Writing – original draft, Project administration, Writing – review and editing;

Shan Zhang, Data curation, Software, Validation, Investigation, Methodology, Writing – original draft, Project administration; Kaiyuan Song, Resources, Software, Formal analysis, Supervision, Investigation, Visualization, Writing – original draft; Luju Jiang, Formal analysis, Validation, Investigation, Methodology, Writing – original draft, Project administration; Li-Peng Hu, Jun Li, Zhi-Wei Cai, Hong-Fei Yao, Investigation, Methodology, Writing – review and editing; Qing Li, Investigation, Methodology, Project administration, Writing – review and editing; Xue-Li Zhang, Investigation, Writing – review and editing; Mingxuan Feng, Chongyi Jiang, Resources, Writing – review and editing; Rong-Kun Li, Formal analysis, Investigation, Methodology, Writing – review and editing; Hui Li, Software, Investigation, Methodology, Writing – review and editing; Jie Chen, Resources, Supervision, Validation, Writing – review and editing; Xiaona Hu, Resources, Supervision, Investigation, Writing – review and editing; Jiaofeng Wang, Resources, Software, Validation, Investigation, Visualization, Writing – review and editing; Helen He Zhu, Visualization, Writing – review and editing; Cun Wang, Validation, Writing – review and editing; Lin-Tai Da, Resources, Supervision, Writing – review and editing; Zhi-Gang Zhang, Conceptualization, Resources, Data curation, Supervision, Validation, Visualization, Writing – review and editing; Zhijun Bao, Conceptualization, Resources, Software, Supervision, Funding acquisition, Validation, Visualization, Writing – review and editing; Xu Wang, Conceptualization, Software, Supervision, Validation, Visualization, Writing – review and editing; Shu-Heng Jiang, Conceptualization, Resources, Data curation, Formal analysis, Supervision, Funding acquisition, Validation, Investigation, Visualization, Methodology, Writing – original draft, Project administration, Writing – review and editing

Author ORCIDs
Shan Zhang ⓘ https://orcid.org/0000-0003-2033-2996
Kaiyuan Song ⓘ https://orcid.org/0000-0001-9889-824X
Rong-Kun Li ⓘ https://orcid.org/0000-0002-0056-2025
Cun Wang ⓘ https://orcid.org/0000-0002-4977-2189
Zhi-Gang Zhang ⓘ https://orcid.org/0000-0003-0423-3381
Shu-Heng Jiang ⓘ https://orcid.org/0000-0001-8516-6234

## Ethics

All animals received proper care in accordance with the guidelines outlined in the 'Guide for the Care and Use of Laboratory Animals' established by the National Academy of Sciences and published by the National Institutes of Health (NIH publication 86-23 revised 1985). The protocols for animal studies were approved by the ethics committee of the Ren Ji Hospital, School of Medicine, Shanghai Jiao Tong University (approval numbers 202201427 and RA-2021-096).

Reviewer #1 (Public review): https://doi.org/10.7554/eLife.103016.4.sa1
Reviewer #2 (Public review): https://doi.org/10.7554/eLife.103016.4.sa2
Author response https://doi.org/10.7554/eLife.103016.4.sa3

# Additional files

## Supplementary files

MDAR checklist

## Data availability

For GSEA analyses, we conducted RNA sequencing (RNA-seq) analysis on HCC-LM3 cells treated with citalopram or fluvoxamine, which led to the identification of 114 differentially expressed genes (DEGs; 80 co-upregulated and 34 co-downregulated), as reported previously (PRJNA1084911). The single-cell transcriptome expression of C5aR1 and GLUT family members in the tumor microenvironment was identified with GSE125449 and GSE140228-10X. All data generated or analyzed during this study are included in the manuscript and supporting files; source data files have been provided for all figures.

The following previously published datasets were used:

| Author(s) | Year | Dataset title | Dataset URL | Database and Identifier |
|---|---|---|---|---|
| Wang XW | 2019 | Tumor cell biodiversity drives microenvironmental reprogramming in liver cancer | https://www.ncbi.nlm.nih.gov/geo/query/acc.cgi?acc=GSE125449 | NCBI Gene Expression Omnibus, GSE125449 |
| Jiang SH | 2024 | RNA sequencing analysis in HCC-LM3 cells upon citalopram or fluvoxamine treatment | https://www.ncbi.nlm.nih.gov/bioproject/PRJNA1084911 | NCBI BioProject, PRJNA1084911 |
| He Y, Zhang Q, Zhang Z, Ren X, Liu K | 2019 | Landscape and Dynamics of Single Immune Cells in Hepatocellular Carcinoma | https://www.ncbi.nlm.nih.gov/geo/query/acc.cgi?acc=GSE140228 | NCBI Gene Expression Omnibus, GSE140228 |

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
