## [Editor Report · eLife Assessment]

This **important** study provides **solid** evidence to support the anti-tumor potential of citalopram, originally an anti-depression drug, in hepatocellular carcinoma (HCC). In addition to their previous report on directly targeting tumor cells via glucose transporter 1 (GLUT1), the authors tried to uncover additional working mechanisms of citalopram in HCC treatment in the current study. The data here suggests that citalopram may regulate the phagocytotic function of TAM via C5aR1 or CD8+T cell function to suppress HCC growth in vivo.

---

## [Referee Report · Reviewer #1 (Public review)]

Summary:

In their previous publication (Dong et al. Cell reports 2024), the authors showed that citalopram treatment resulted in reduced tumor size by binding to the E380 site of GLUT1 and inhibiting the glycolytic metabolism of HCC cells, instead of the classical citalopram receptor. Given that C5aR1 was also identified as the potential receptors of citalopram in the previous report, the authors focused on exploring the potential of immune-dependent anti-tumor effect of citalopram via C5aR1. C5aR1 was found to be expressed on tumor-associated macrophages (TAMs) and citalopram administration showed potential to improve the stability of C5aR1 in vitro. Through macrophage depletion and adoptive transfer approaches in HCC mouse models, the data demonstrated the potential importance of C5aR1-expressing macrophage in the anti-tumor effect of citalopram in vivo. Mechanistically, their data suggested that citalopram may regulate the phagocytosis potential and polarization of macrophages through C5aR1, thereby potentiated CD8+T cell responses in vivo. Finally, as the systemic 5-HT level is down-regulated by citalopram, the authors analyzed the association between a low 5-HT and a superior CD8+T cell function against tumor.

Strengths:

The idea of repurposing clinical-in-used drugs showed great potential for immediate clinical translation. The data here suggested that the anti-depression drug, citalopram displayed immune regulatory role on TAM via a new target C5aR1 in HCC.

Comments on revised version:

The authors have already addressed the previous comments.

---

## [Referee Report · Reviewer #2 (Public review)]

Summary:

Dong et al. present a thorough investigation into the potential of repurposing citalopram, an SSRI, for hepatocellular carcinoma (HCC) therapy. The study highlights the dual mechanisms by which citalopram exerts anti-tumor effects: reprogramming tumor-associated macrophages (TAMs) toward an anti-tumor phenotype via C5aR1 modulation and suppressing cancer cell metabolism through GLUT1 inhibition, while enhancing CD8+ T cell activation. The findings emphasize the potential of drug repurposing strategies and position C5aR1 as a promising immunotherapeutic target.

Strength:

It provides detailed evidence of citalopram's non-canonical action on C5aR1, demonstrating its ability to modulate macrophage behavior and enhance CD8+ T cell cytotoxicity. The use of DARTS assays, in silico docking, and gene signature network analyses offers robust validation of drug-target interactions. Additionally, the dual focus on immune cell reprogramming and metabolic suppression presents a comprehensive strategy for HCC therapy. By highlighting the potential of existing drugs like citalopram for repurposing, the study also underscores the feasibility of translational applications. During revision, the authors experimentally demonstrated that TAM has lower GLUT1 levels, further strengthening their claim of C5aR1 modulation-dependent TAM improvement for tumor therapy.

Comments on revised version:

The authors have addressed most of my concerns about the paper.

---

## [Author Response]

The following is the authors’ response to the previous reviews.

**Public Reviews:**

**Reviewer #1 (Public review):**
Summary:In their previous publication (Dong et al. Cell Reports 2024), the authors showed that citalopram treatment resulted in reduced tumor size by binding to the E380 site of GLUT1 and inhibiting the glycolytic metabolism of HCC cells, instead of the classical citalopram receptor. Given that C5aR1 was also identified as the potential receptors of citalopram in the previous report, the authors focused on exploring the potential of immune-dependent anti-tumor effect of citalopram via C5aR1. C5aR1 was found to be expressed on tumor-associated macrophages (TAMs) and citalopram administration showed potential to improve the stability of C5aR1 in vitro. Through macrophage depletion and adoptive transfer approaches in HCC mouse models, the data demonstrated the potential importance of C5aR1-expressing macrophage in the anti-tumor effect of citalopram in vivo. Mechanistically, their in vitro data suggested that citalopram may regulate the phagocytosis potential and polarization of macrophages through C5aR1. Next, they tried to investigate the direct link between citalopram and CD8+T cells by including an additional MASH-associated HCC mouse model. Their data suggest that citalopram may upregulate the glycolytic metabolism of CD8+T cells, probability via GLUT3 but not GLUT1-mediated glucose uptake. Lastly, as the systemic 5-HT level is down-regulated by citalopram, the authors analyzed the association between a low 5-HT and a superior CD8+T cell function against tumor. Although the data is informative, the rationale for working on additional mechanisms and logical link among different parts are not clear. In addition, some of the conclusion is also not fully supported by the current data.Strengths:The idea of repurposing clinical-in-used drugs showed great potential for immediate clinical translation. The data here suggested that the anti-depression drug, citalopram displayed immune regulatory role on TAM via a new target C5aR1 in HCC.Comments on revised version:The authors have addressed most of my concerns about the paper.

We thank you the reviewer. We appreciate the reviewer’s constructive suggestions that helped improve the clarity and robustness of the study.

**Reviewer #2 (Public review):**
Summary:Dong et al. present a thorough investigation into the potential of repurposing citalopram, an SSRI, for hepatocellular carcinoma (HCC) therapy. The study highlights the dual mechanisms by which citalopram exerts anti-tumor effects: reprogramming tumor-associated macrophages (TAMs) toward an anti-tumor phenotype via C5aR1 modulation and suppressing cancer cell metabolism through GLUT1 inhibition, while enhancing CD8+ T cell activation. The findings emphasize the potential of drug repurposing strategies and position C5aR1 as a promising immunotherapeutic target.Strengths:It provides detailed evidence of citalopram's non-canonical action on C5aR1, demonstrating its ability to modulate macrophage behavior and enhance CD8+ T cell cytotoxicity. The use of DARTS assays, in silico docking, and gene signature network analyses offers robust validation of drug-target interactions. Additionally, the dual focus on immune cell reprogramming and metabolic suppression presents a comprehensive strategy for HCC therapy. By highlighting the potential for existing drugs like citalopram to be repurposed, the study also emphasizes the feasibility of translational applications. During revision, the authors experimentally demonstrated that TAM has lower GLUT1, which further strengthens their claim of C5aR1 modulation-dependent TAM improvement for tumor therapy.Weaknesses:The authors proposed that CD8+ T cells have an TAM-independent role upon Citalopram treatment. However, this claim requires further investigation to confirm that the effect is truly "TAM independent".

We appreciate the reviewer’s insightful comment regarding the interpretation of CD8^+^ T cell roles. In this study, in vitro analyses show that citalopram directly enhances CD8^+^T cell activity, as evidenced by increased CFSE proliferation, upregulation of activation markers, and cytotoxic effector readouts (Figures S10A–E). Accordingly, we infer a TAM-independent CD8^+^ T cell activation by citalopram in vitro.

Our in vivo data indicate that the primary anti-tumor mechanism of citalopram involves targeting C5aR1^+^ TAMs, which subsequently enhances CD8^+^ T cell immunity. This conclusion is supported by the near-complete ablation of citalopram’s therapeutic effect upon TAM depletion with clodronate liposomes (Figure S5). Additionally, citalopram reduces serum serotonin (5-HT) levels (Figure 4E), recapitulating the serotonergic state of Tph1^−/−^ mice. Notably, the anti-tumor effect and CD8^+^ T cell activation induced by citalopram exceed those observed in Tph1^−/−^ mice (Figures 4G–I), suggesting that 5-HT reduction contributes to CD8^+^ T cell activation but operates alongside other mechanisms in vivo, prominently including TAM targeting. As suggested, we further tested CD8^+^ T cell activity in the context of macrophage depletion. The result showed that citalopram did not further enhance CD8^+^ T cell cytotoxicity after macrophage depletion, indicating that TAM-dependent pathways are central to CD8^+^ T cell–mediated anti-tumor immunity and largely underlie the anti-tumor effects of citalopram.

To accurately reflect our main findings, we had made several revisions to the manuscript. First, we have revised the title to “Citalopram exhibits immune-dependent anti-tumor effects by modulating C5aR1^+^ TAMs”. In the Results section, the Conclusions have been updated to: “These data not only corroborate recent reports that SSRIs modulate CD8^+^ T cell function via serotonergic-dependent mechanism, but also reveals additional in vivo regulatory avenues by which citalopram affects CD8^+^ T cells, such as its ability to reprogram C5aR1^+^ TAMs. Notably, in the context of macrophage depletion, CD8^+^ T cell cytotoxicity was not further enhanced by citalopram, indicating that TAM-dependent pathways are central to CD8^+^ T cell-mediated anti-tumor immunity and largely underlie the anti-tumor effects of citalopram”. In the Discussion part, we have included the following content: “Although citalopram directly stimulates CD8^+^ T cells in vitro, the TAM-independent activation is not evident in vivo within the complex TME, as CD8^+^ T cell responses are abolished by macrophage depletion, indicating that the in vivo effects of citalopram on CD8^+^ T cells and tumor growth are largely TAM-dependent”.

**Recommendations for the authors:**

**Reviewer #2 (Recommendations for the authors):**
Fig S5 and Fig 3: To improve clarity regarding the roles of TAMs and CD8+ T cells, can the authors experimentally demonstrate the macrophage-independent function of CD8+ T cells? An experiment in Fig 3J using or not using Clodro-Liposome to deplete TAMs would be more informative.

We thank the reviewer for the insightful suggestion. In this study, in vitro analyses show that citalopram directly enhances CD8^+^ T cell activity, as evidenced by increased CFSE proliferation, upregulation of activation markers, and cytotoxic effector readouts (Figures S10A–E). Therefore, we conclude a TAM-independent CD8^+^ T cell activation induced by citalopram. Previously, in Figure S5, we analyzed the therapeutic effect of citalopram after macrophage depletion by clodronate liposomes and also probed the immune profiles. The result showed that CD8^+^ T cell cytotoxic activities were not significantly affected by citalopram in this context (Figure S5E), indicating that the TAM-dependent pathway is central to CD8^+^ T cell-mediated anti-tumor immunity and to the anti-tumor effects of citalopram. We have incorporated this result into the revised manuscript.

Fig S4: The figure panel showing sample/treatment annotations is missing.

Thank you for pointing this out. We have updated Fig. S4 to include explicit sample identifiers, treatment group labels, and drug concentrations.

Since Glut3 is vital in both TAMs and CD8+ T cells, the authors should discuss the interaction between Glut3 and Citalopram. Additionally, include details about the structural homology between Glut1 and Glut3 in the discussion.

Thank you for the suggestion. Citalopram was docked into the GLUT1 substrate-binding pocket, with the best poses showing an electrostatic interaction centered on E380 accompanied by hydrophobic contacts within the pocket (Our previous publication, Dong et al. Cell Reports 2024). Although GLUT1 and GLUT3 share a highly conserved core substrate-binding pocket, isoform-specific regulation arises from features outside the canonical site. Structural homology between GLUT1 and GLUT3 is high in the transmembrane core, but regulatory features, such as the cytosolic Sugar Porter (SP) motif network, the conserved A motif, lipid interfaces, and gating dynamics, differ between the two isoforms (PMID: 33536238). These regulatory differences can alter pocket accessibility, coupling to conformational transitions, and allosteric communication with the cytosol, such that a ligand binding GLUT1 in the inward-facing state may not stabilize a GLUT3 conformation that yields appreciable transport inhibition. Consistently, functional experiments have indicated robust GLUT1 engagement in cancer cells (Dong et al. Cell Reports 2024), while equivalent GLUT3 inhibition has not been observed in TAMs (Figure S8), suggesting isoform-selective targeting by citalopram. We have included these discussion in the revised manuscript.

Fig 3O: Please clarify the statement regarding the requirements of CD8 T cells for the pro-tumor phenotype of C5aR1+ TAMs. Specify whether this relates to a pro- or anti-tumor effect of CD8 T cells.

Thanks. As suggested, we have improved the statement as follows: “depletion of CD8^+^ T cells abrogated the C5aR1^+^ TAM-mediated enhancement of tumor growth (Figure 3O), suggesting that the anti-tumor effects of CD8^+^ T cells are required for the pro-tumor phenotype of C5aR1^+^ TAMs”.